# ProSAR: Prototype-Guided Semantic Augmentation and Refinement for Time Series Contrastive Learning

Caiyi Yang [1]   Chenglin Li [1]   Hao Zhang [2]   Weijia Lu [3]   Zhifei Yang [3]   Wenrui Dai [1]   Xiaodong Zhang [3]
Xiaofeng Ma [3]   Can Zhang [3]   Junni Zou [1]   Hongkai Xiong [1]

## Abstract

Contrastive learning has advanced the representation learning across domains, yet its success relies on data augmentations that preserve semantic contents while providing the view diversities. Multivariate time series, however, are inherently noisy, non-stationary, and lack such intuitive semantic cues. Consequently, standard heuristic augmentations that ignore semantic parts may risk destroying critical temporal dependencies. Though some recent approaches attempt to isolate informative components, they typically rely on an implicit neural mechanism to infer semantics, thus limiting the interpretability and controllability. To address this, we propose ProSAR, an information-theoretic framework that leverages the explicit prototype alignment to guide semantic augmentations, and establish a feedback loop between the augmentation, contrastive learning, and prototype updates. Specifically, grounded in our proposed Prototype-Conditioned Information Bottleneck principle, we leverage the time-domain prototypes as explicit anchors to localize semantic segments, and develop a time–frequency augmentation strategy that retains prototype-consistent information while discarding noise. To promote semantically consistent prototypes for a reliable view generation, we design a dual-prototype loop where the augmented views are encoded into representations and then the learned representations are clustered to update latent prototypes, whose decoded feedback refines the time-domain prototypes for the next round of augmentation. Experiments on diverse time-series benchmarks demonstrate that

ProSAR outperforms the other contrastive learning methods on downstream forecasting and classification tasks.

## 1. Introduction

The proliferation of time series data across diverse domains, from healthcare (Miotto et al., 2016), finance (Heaton et al., 2017) to industrial IoT (Syafrudin et al., 2018) and human activity recognition (Wang et al., 2019), has underscored the imperative need for an analytical tool for the effective representation learning. However, the high costs and numerous efforts associated with manual labeling often render the commonly-used supervised learning impractical, motivating the rapid development of self-supervised learning (SSL) paradigms (Jaiswal et al., 2020; Misra & Maaten, 2020). Among them, contrastive learning (CL) has emerged as a particularly promising approach (Le-Khac et al., 2020; Chen et al., 2020; He et al., 2020), which aims to learn the representations by maximizing the agreement between different views of the same data instance while minimizing the agreement with views of the other data instances, where the views are typically generated through data augmentation (Shorten & Khoshgoftaar, 2019). Alongside some direct efforts to improving augmentation, the field has advanced on parallel fronts—refining the contrastive objective (Lee et al., 2024), exploiting the multi-frequency structure (Duan et al., 2024), and modeling the relative similarity (Xu et al., 2025), yet the view-generation module itself remains a fundamental bottleneck.

Despite the success of CL, its efficacy on the time series data is profoundly challenged by the difficulty of designing data augmentations that can preserve some crucial temporal semantics, while ensuring a sufficient view diversity (Wen et al., 2021; Luo et al., 2023). Unlike images or texts, where human intuition can often guide a semantic-preserving transformation, the complex, often non-intuitive, temporal structures in time series make the augmentation design a formidable task (Zheng et al., 2024). Many existing augmentation methods are either based on the hand-picked heuristics, e.g., jittering, scaling, permutation (Zhang et al.,

[1]Shanghai Jiao Tong University, Shanghai, China. [2]East China Normal University, Shanghai, China. [3]United Automotive Electronic Systems, Shanghai, China. Correspondence to: Chenglin Li <LCL1985@sjtu.edu.cn>, Hao Zhang <hzhang@cee.ecnu.edu.cn>, Junni Zou <zoujunni@sjtu.edu.cn>.

*Proceedings of the 43rd International Conference on Machine Learning*, Seoul, South Korea. PMLR 306, 2026. Copyright 2026 by the author(s).

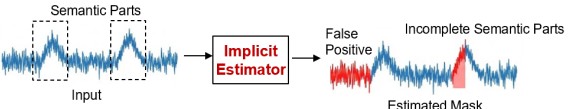

*(a)* Existing semantic-aware methods (e.g., AutoTCL)

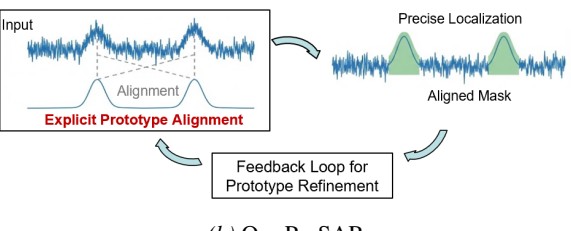

*(b)* Our ProSAR

*Figure 1.* Comparison between existing semantic-aware methods and ours. (a) Existing implicit methods often suffer false positives and incomplete masks. (b) Our ProSAR utilizes the explicit prototypes and an additional feedback loop for a precise localization.

2022; Yue et al., 2022; Eldele et al., 2021), or a direct adaptation from the other modalities, at the risk of incurring distortion or destruction of the pivotal temporal patterns and semantic information (Tian et al., 2025). This cross-modality incompatibility may lead to mismatched patterns and even inadvertent generation of false negative pairs, thus hindering the learning of a robust representation (Meng et al., 2023; Chuang et al., 2020).

To address this, current CL methodologies for time series has strived for the semantic augmentation from an information-theoretic perspective. Theoretical groundwork like the InfoMin principle (Tian et al., 2020) posits that desirable views reduce the mutual information while preserving the task-relevant signals. InfoTS (Luo et al., 2023) extends InfoMin with a criteria to balance between the augmentation fidelity and variety, by further introducing a meta-learner for the adaptive selection. Frameworks such as AutoTCL (Zheng et al., 2024) learn to factorize the instances for augmentation, by separating the informative parts from the task-irrelevant ones. Additionally, FreRA (Tian et al., 2025) learns the frequency-importance, by preserving the critical subbands while perturbing non-critical ones to produce the semantics-preserving views. Nonetheless, despite these advances in the semantic-aware augmentation, the preserved content is still learned and inferred indirectly, rather than tied to the explicit and temporally aligned anchors, thus limiting the interpretability and controllability.

We are thus motivated to propose in this paper a contrastive learning framework for time series, named ProSAR, i.e., prototype-guided semantic augmentation and refinement. ProSAR's core innovation is to turn the semantic view generation from a black-box heuristic into an explicit, anchor-aligned policy that is continuously improved by a prototype-refinement feedback loop, within our information-theoretic

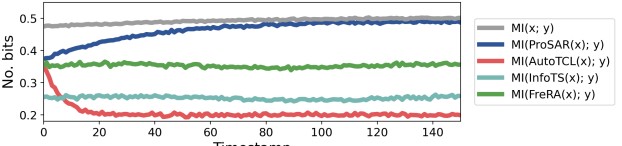

*Figure 2.* Mutual information results between the generated views and the label across timestamps. Higher values indicate stronger retention of label-relevant information.

principle. Theoretically, we formalize semantic augmentation through the information bottleneck principle (Tishby et al., 2000), positing that desirable views should retain information consistent with their associated semantic prototype while discarding prototype-irrelevant variations. Guided by this principle, we introduce learnable time-domain prototypes as the explicit anchors to align each series and localize semantically coherent segments, and then perform a time-frequency augmentation on these segments to construct views that satisfy our information-theoretic objective. Moreover, motivated by observations that in the latent space, prototypes obtained via clustering capture higher-level semantics (Li et al., 2021a; To et al., 2025), we design a dual-prototype refinement loop in which latent-space prototypes, updated from clustering the learned representations, provide decoded feedback to update the time-domain prototypes for the next round of localization and augmentation.

As shown in Fig. 1, ProSAR differs from the existing methods in how semantic segments are localized for view generation. Existing semantic-aware augmentations (Zheng et al., 2024; Tian et al., 2025) typically infer the semantic regions implicitly via learned masks, which can be inaccurate or incomplete (Fig. 1a). In contrast, our ProSAR (Fig. 1b) leverages prototypes as explicit anchors and localizes semantic segments through alignment, offering a more controllable and interpretable basis for view generation. Moreover, ProSAR introduces a prototype-refinement loop where the aligned semantic segments provide feedback to update the prototypes. To provide an information-theoretic perspective, Fig. 2 further reports the mutual information (MI) between the augmented view $v$ and the downstream label $y$ in training, where $v$ is generated from the input series $x$ by ProSAR and the other baselines. A higher $\mathrm{MI}(v; y)$ indicates that $v$ preserves more label-relevant semantic information. Compared to AutoTCL (Zheng et al., 2024) and FreRA (Tian et al., 2025) which infer the semantic regions implicitly via masks, and InfoTS (Luo et al., 2023) which selects augmentations via MI without an explicit region inference, ProSAR achieves the highest $\mathrm{MI}(v; y)$ throughout training. Moreover, $\mathrm{MI}(\mathrm{ProSAR}(x); y)$ steadily approaches $\mathrm{MI}(x; y)$, benefiting from our prototype-refinement feedback loop. We provide more details regarding Fig. 2 in Appendix C.7. Our main contributions are as follows.

- We propose ProSAR, an information-theoretic framework that leverages an explicit prototype alignment to guide

semantic augmentations, and establishes a feedback loop between the augmentation, contrastive learning, and prototype updates.

- We formalize the semantic augmentation via a Prototype-Conditioned Information Bottleneck principle, which yields an explicit prototype-optimal augmentation criterion. We further provide a theoretical analysis for the resulting prototype–augmentation co-design loop.

- We leverage the learnable time-domain prototypes as explicit anchors to align time series and localize semantically coherent segments, and then perform a time-frequency augmentation on these segments to construct views aligned with our information-theoretic objective.

- We introduce a dual-prototype refinement loop that feeds back the latent-space clustered prototypes to iteratively update the time-domain prototypes, improving semantic localization and view generation for contrastive learning.

## 2. Related Work

### 2.1. Self-Supervised and Contrastive Representation Learning for Time Series

Early works on SSL demonstrate that temporal structure can be learned without labels via predictive or contrastive objectives, e.g., CPC (Oord et al., 2018) and TLoss (Franceschi et al., 2019). For time series, representative methods exploit the temporal neighborhoods (TNC) (Tonekaboni et al., 2021), multi-level context and subseries consistency (TS2Vec) (Yue et al., 2022), disentangled seasonal–trend factors (CoST) (Woo et al., 2022), and Transformer encoders for multivariate sequences (TST) (Zerveas et al., 2021). For vision data, SimCLR (Chen et al., 2020) and MoCo (He et al., 2020) catalyze the modern contrastive learning. Recent time-series advances include SoftCLT, which uses soft assignments at instance and timestamp levels (Lee et al., 2024), and MF-CLR, which imposes cross-frequency consistency with a hierarchical mechanism (Duan et al., 2024). TimesURL incorporates the frequency-temporal augmentation, hard negatives, and joint reconstruction (Liu & Chen, 2024). PPT emphasizes that patch order matters for time series, and introduces a patch-order prediction pretext task to learn temporally aware representations (Kim et al., 2025).

### 2.2. Augmentation for Contrastive Learning

Constructing views that are both diverse and semantics-preserving is critical to contrastive learning (Tian et al., 2020). The survey of time-series augmentation (Wen et al., 2021) summarizes common hand-crafted transforms and analyzes their limitations. TS-TCC adopts heuristic policies, using jitter and scaling as weak transformations and permutation with jitter as strong ones (Eldele et al., 2021). TF-C promotes the view quality by aligning time- and frequency-

domain representations during pretraining, encouraging consistency across complementary domains (Zhang et al., 2022). From an information-theoretic perspective, InfoMin prescribes reducing mutual information between views while retaining task-relevant content (Tian et al., 2020). Building upon this, InfoTS scores the candidate transforms and adaptively selects them to balance between the fidelity and diversity (Luo et al., 2023). AutoTCL factorizes each instance into the informative and task-irrelevant components, and learns parametric augmenters to target these parts (Zheng et al., 2024). A frequency-aware line further learns the band importance: FreRA preserves the critical bands while perturbing non-critical ones to form semantics-preserving views (Tian et al., 2025). Beyond augmentation itself, AutoCL automatically searches for the data augmentations, embedding transformations, contrastive pair construction, and contrastive losses (Jing et al., 2024).

### 2.3. Prototype- and Cluster-Aware Contrastive Learning

Incorporating prototypes or cluster structure into the objective can inject semantic priors beyond the instance discrimination. For example, PCL introduces ProtoNCE to pull samples towards the assigned prototypes (Li et al., 2021a). SwAV contrasts cluster assignments via online clustering and assignment consistency (Caron et al., 2020). Contrastive clustering jointly performs the instance- and cluster-level contrast (Li et al., 2021b), and the graph contrastive clustering extends the cluster-aware contrast to graphs (Zhong et al., 2021). For time series, MHCCL uses the hierarchical clustering with downward/upward masking to refine prototypes and perform cluster-wise contrast (Meng et al., 2023). AimTS proposes a two-level prototype-based contrast with series–image cross-modal contrast to better leverage existing augmentations, but does not use the prototypes to drive augmentation itself (Chen et al., 2025). Our ProSAR differs by elevating the learnable prototypes to explicit semantic carriers that guide upstream view generation under the information-theoretic principle. Departing from the prior augmentation strategies, ProSAR utilizes prototypes as the explicit guides for a more interpretable view generation process. Moreover, extending beyond prototypical methods that use prototypes solely at the objective level, ProSAR implements a closed-loop mechanism, where these prototypes are dynamically refined to steer the augmentation policy.

## 3. Proposed Method

In this section, we introduce ProSAR, a self-supervised framework for learning semantically rich representations of time series, which distinctively incorporates learnable prototypes into the information-theoretic design of data augmentations. We first establish the theoretical principles in Section 3.1, demonstrating how prototypes can guide the

creation of semantically consistent views and how, in turn, these views are used to refine the prototypes. Subsequently, Section 3.2 details the architectural components and learning objectives that implement this co-design, culminating in a robust and theoretically grounded approach to time series representation learning. For clarity, the key notations employed throughout this paper are summarized in Table 1.

### 3.1. Information-Theoretic Foundations for Prototype-Enhanced Semantic Augmentation

We begin by formalizing semantic data augmentation through the lens of information theory. Our objective is to generate the augmented views $\tilde{X}$ of an input time series $X$ that preserve its essential semantic content represented by a latent variable $C$, while discarding irrelevant information. A core challenge in self-supervised learning is that $C$ is unknown. This motivates our introduction of the learnable prototypes $\mathbf{P}$ (specifically, the latent prototypes $\{p_k^z\}$) as tractable proxies for $C$, enabling a co-design of the augmentation strategies and prototype refinement.

#### 3.1.1. Information Bottleneck and its Challenge in Self-Supervised Learning

We let $X$ denote a random variable representing a raw time series, and $C$ be a latent variable encapsulating its core semantic information. An augmentation $T$ generates an augmented view $\tilde{X} = T(X)$. The design of $T$ is guided by the information bottleneck (IB) principle, which employs mutual information (MI) to quantify dependence between variables. The principle formalizes this objective as a trade-off: maximizing the MI with the latent semantics $I(C; \tilde{X})$ to retain essential meaning, while simultaneously minimizing the MI with the input $I(X; \tilde{X})$ to achieve compression:

$$\max_T I(C; \tilde{X}) - \beta I(X; \tilde{X}), \qquad (1)$$

where $\beta > 0$ balances semantic fidelity and compression. This aligns with the InfoMin principle for contrastive learning (Tian et al., 2020). In SSL, $C$ is unknown, rendering a direct optimization of Eq. (1) infeasible and thus necessitating a data-driven proxy for $C$.

#### 3.1.2. Prototypes as Semantic Proxies and Conditional Information Bottleneck

To address the unknown $C$, we introduce $K$ learnable prototypes, $\mathbf{P}$, representing distinct semantic clusters (see Table 1). A prototype assignment variable $P \in \{1, \ldots, K\}$ indicates the semantic cluster most associated with an input time series $X$. In our framework, this assignment is operationalized in the input space: $P$ is determined by identifying the time-domain prototype $p_k^x$ that best aligns with $X$. Thus, $P$ remains a deterministic function of $X$ (i.e., $P = \mathrm{assign}(X, \{p_k^x\})$). Substituting $P$ for $C$ in Eq. (1) yields the *prototype-conditioned IB objective* for augmenta-

Table 1. Key notations, where RV stands for a random variable.

| Symbol | Description |
|---|---|
| $X, x$ | Time series (RV, instance) |
| $\tilde{X}, \tilde{x}$ | Augmented views of $X, x$ |
| $C$ | Latent semantic variable |
| $\mathbf{P} = \{p_k^z\}$ | Set of $K$ latent prototypes |
| $P$ | Prototype assignment index |
| $f_\theta$ | Encoder network |
| $z$ | Latent representation vector |
| $p_k^x$ | Time-domain prototype |
| $p_k^z$ | Latent prototype |
| $x_S, x_N$ | Semantic, non-semantic parts of $x$ |
| $M_x$ | Binary mask on instance $x$ |
| $I(A; B)$ | Mutual Information |
| $\beta$ | IB objective hyperparameter |
| $\mathcal{L}_{\text{total}}$ | Overall loss function |
| $T$ | Augmentation transformation |
| $D_\psi$ | Decoder network |

tion design:

$$\max_T I(P; \tilde{X}) - \beta I(X; \tilde{X}). \qquad (2)$$

This links augmentation design to the learned prototypes, setting the stage for their co-design. Compared with PCL (Li et al., 2021a) that uses prototypes at the objective level, ProSAR distinctively leverages these prototypes to actively guide the augmentation generation process itself through an information-theoretic lens.

To analyze Eq. (2), we first decompose $I(X; \tilde{X})$ by using the fact that $P$ is a function of $X$.

**Proposition 3.1.** *If $P$ is determined by $X$ (i.e., $P = g(X)$ for some function $g$), then $I(X; \tilde{X}) = I(P; \tilde{X}) + I(X; \tilde{X} \mid P)$.*

Substituting this decomposition into Eq. (2), the objective becomes:

$$\begin{aligned} \max_T & I(P; \tilde{X}) - \beta(I(P; \tilde{X}) + I(X; \tilde{X} \mid P)) \\ &= \max_T (1 - \beta) I(P; \tilde{X}) - \beta I(X; \tilde{X} \mid P). \end{aligned} \qquad (3)$$

This transformed objective leads to the following characterization of an optimal augmentation strategy with respect to the current prototype assignments $P$.

**Proposition 3.2** (Prototype-Optimal Augmentation). *For $\beta \in (0, 1)$, an augmentation $T^*$ optimizing Eq. (3) aims to satisfy:*

*(i) $I(X; \tilde{X} \mid P) \to 0$: $\tilde{X}$ retains minimal $X$-information not explained by $P$;*

*(ii) $I(P; \tilde{X}) \to I(P; X)$: $\tilde{X}$ is maximally informative about $P$.*

Please refer to Appendix E for the detailed proof. Proposition 3.2 implements the InfoMin principle (Tian et al.,

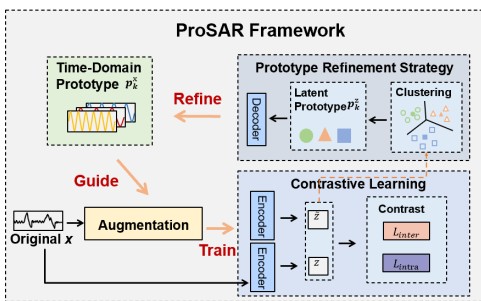

*Figure 3.* Overall training framework of ProSAR.

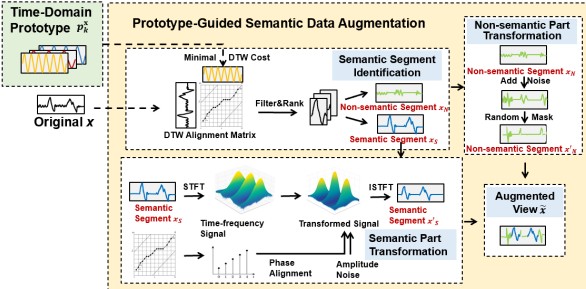

*Figure 4.* Prototype-guided semantic data augmentation.

2020) by using the prototypes. It implies that as prototype assignments $P$ (derived from $f_\theta$ and $\mathbf{P}$) evolve, the optimal augmentation $T^*$ must also co-evolve. Condition (*i*) requires the augmentation to discard information within $X$ that is irrelevant to the prototype assignment $P$, thereby isolating the informative part from the task-irrelevant part. Concurrently, Condition (*ii*) ensures that $\tilde{X}$ remains maximally informative about $P$, preserving the core semantic signal. However, ProSAR explicitly defines this part through its relevance to the learned prototypes $\mathbf{P}$.

### 3.1.3. Co-design and Iterative Refinement of Prototypes and Augmentations

Eq. (3) underscores the critical interdependency for co-design: the optimal augmentation $T$ depends on the quality of the time-domain prototypes $\{p_k^x\}$ that determine the assignment $P$, while these prototypes are themselves refined by learning from the augmented views. This creates a synergistic and iterative process.

**Data Augmentation Guided by Time-Domain Prototypes:** Proposition 3.2 indicates that augmentations should make the views highly informative about their assigned prototype $P$, while discarding $P$-irrelevant information from $X$. Since $P$ is determined by alignment with time-domain prototypes $\{p_k^x\}$, this motivates us to transform $X$ based on the segments that are semantically relevant to these time-domain anchors.

**Dual Prototype Refinement via Augmentations:** The refinement of the time-domain prototypes $\{p_k^x\}$ is achieved through a synergistic process involving the encoder $f_\theta$ and a corresponding set of latent prototypes $\{p_k^z\}$. Specifically, the encoder learns from augmented views to produce robust latent representations $(z, \tilde{z})$. These representations are then used to update the latent prototypes $\{p_k^z\}$ to better capture high-level semantic clusters. In turn, these refined latent prototypes guide the evolution of the time-domain prototypes $\{p_k^x\}$, creating an indirect but powerful refinement pathway.

This interplay between time-domain guidance and latent-space learning forms a positive feedback loop: *i*) better time-domain prototypes yield more semantically aligned augmentations; *ii*) these augmentations produce more con-

sistent views, enabling the encoder to learn more discriminative latent representations, which are then clustered to refine the latent prototypes; and *iii*) the refined latent prototypes, via decoded feedback, facilitate the update of higher-quality time-domain prototypes. Formally, under an idealized alternating-update scheme, we show that the expected objective admits a monotone improvement property and thus converges.

**Theorem 3.3** (Convergence of Co-design Loop). *Let $\mathcal{J}(P, T)$ be the expected prototype-conditioned IB objective. The alternating updates of prototype assignment $P$ and augmentation policy $T$ constitute a block coordinate ascent, satisfying the monotonic improvement property:*

$$\mathcal{J}(P^{(t+1)}, T^{(t+1)}) \geq \mathcal{J}(P^{(t)}, T^{(t)}), \qquad (4)$$

*which implies the convergence to a limit $\lim_{t\to\infty} \mathcal{J}(P^{(t)}, T^{(t)}) = \mathcal{J}^*$.*

*Proof.* See Appendix E for detailed proof. $\square$

### 3.2. ProSAR: Framework and Implementation

Building upon the information-theoretic principles of co-design and iterative refinement from Section 3.1, we implement ProSAR in a practical framework as shown in Fig. 3. At each iteration, given an input series $x$, the current prototypes guide the augmentation module to generate a semantically aligned view $\tilde{x} = T(x)$, and the encoder maps $(x, \tilde{x})$ to representations $(z, \tilde{z})$ for contrastive training. The updated representations are then clustered to refine prototypes for the next iteration, forming a closed loop between augmentation, contrastive learning and prototype updates. The detailed Algorithm 1 is provided in Appendix A.

#### 3.2.1. Prototype-Guided Semantic Data Augmentation

To implement Proposition 3.2 without access to the true $C$, we employ the learnable *time-domain prototype sequences* $\{p_k^x\}$. These act as semantic anchors in the input space, whose learning and connection to latent prototypes $p_k^z$ are detailed in Section 3.2. As illustrated in Fig. 4, we first align $x$ to its closest time-domain prototype to identify a

semantic mask $M_x$, and then apply a time-frequency transformation to the semantic part $x_S$ and strong perturbations to the non-semantic part $x_N$ to form the final view $\tilde{x}$.

(a) **Semantic Segment Identification:** Multiple informative segments, denoted as a set $\{S_{\text{raw},j}\}$, are identified within the input time series $x$. This process leverages the alignment of $x$ with relevant time-domain prototypes $\{p_k^x\}$, computed using soft-Dynamic Time Warping (soft-DTW) (Cuturi & Blondel, 2017). For the input time series $x$, we first identify its best-matching time-domain prototype via soft-DTW. Subsequently, sub-segments of $x$ with a high-quality alignment to this specific prototype are considered candidate semantic segments. These candidates are then typically filtered based on their alignment scores to select the most relevant segments. To further refine this selection by explicitly seeking portions of $x$ most semantically related to the concepts embodied by the prototypes, a measure of mutual information between a candidate segment and its guiding prototype can be maximized. In practice, we estimate this mutual information using the same kNN-based nonparametric estimator as Appendix C.7. A binary mask $M_x$ is then defined, where $M_x(t) = 1$ if time step $t$ falls within any of the selected semantic segments $S_{\text{raw},j}$, and $M_x(t) = 0$ otherwise. This mask identifies the semantic part $x_S = x \odot M_x$ and the non-semantic part $x_N = x \odot (1 - M_x)$. This entire step utilizes prototypes to estimate the informative parts of $x$ relevant to its underlying semantic structure as captured by the prototype assignments $P$.

(b) **Transformation:** Once $x_S$ and $x_N$ are defined, the augmented view $\tilde{x}$ is constructed by transforming these parts to satisfy the conditions of Proposition 3.2.

1. **Transform Semantic Part ($x_S$):** The identified semantic part $x_S$ undergoes the following two transformations to produce $x_S'$.
- *Temporal Alignment in the Frequency Domain:* The alignment path between $x_S$ and the guiding $p_k^x$ reveals local temporal misalignments. This information is used to apply phase compensation in the frequency domain. Identified local time shifts can be compensated by adjusting the phase of the STFT representation of $x_S$. This aims to normalize temporal variations, preserving semantic integrity and helping to ensure $I(P; x_S') \approx I(P; x_S)$.
- *Controlled Noise Injection:* Further, controlled noise, specifically by perturbing frequency domain amplitudes, is applied to the temporally-aligned segment. $x_S'$ is obtained after this step, where the noise helps in reducing superficial information in $x_S$ not essential for identifying $P$, thereby contributing to reducing $I(X; \tilde{X} \mid P)$. Let the result of the two transformations on $x_S$ be $x_S'$.

2. **Perturb Non-Semantic Part ($x_N$):** The non-semantic part $x_N$ is heavily modified to produce $x_N'$. This involves first applying strong noise perturbation to $x_N$, followed by

random sub-segment masking within $x_N$. This two-fold process aims to thoroughly corrupt information in $x_N$ that is irrelevant to $P$ for these non-semantic regions.

3. **Construct Augmented View $\tilde{x}$:** The final augmented view is assembled by combining the transformed semantic part and the perturbed non-semantic part: $\tilde{x} = (x_S' \odot M_x) + (x_N' \odot (1 - M_x))$. Effectively, $\tilde{x}$ largely preserves the prototype-relevant semantics from $x_S$ while minimizing other information from $x_S$ and heavily disrupting $x_N$.

This adaptive process, which precisely manipulates the prototype-relevant part $x_S$ and the irrelevant part $x_N$ based on guidance from $p_k^x$, is a direct attempt to instantiate the optimal augmentation strategy. The nature and learning of these time-domain prototypes $p_k^x$ and their intrinsic link to latent prototypes $p_k^z$ are crucial for semantic consistency and are further detailed below.

### 3.2.2. Prototype Refinement Strategy

As illustrated in the refinement module of Fig. 3, ProSAR updates prototypes in a closed loop where clustering the learned representations refines the latent prototypes $p_k^z$, which are then decoded to update the time-domain prototypes $p_k^x$ that guide the next-round augmentation. Dynamic prototype updates are crucial for capturing evolving semantic structures. The primary mechanism for refining latent prototypes $p_k^z$ is Latent-Space Clustering, aligning with the iterative improvement principles in Section 3.1. We employ the FINCH (Sarfraz et al., 2019) hierarchical clustering algorithm, applied to a combined set of instance representations from original views $\{z_i = f_\theta(x_i)\}$ and their corresponding augmented views $\{\tilde{z}_i = f_\theta(\tilde{x}_i)\}$. Following clustering, a set of representative cluster centroids is computed. These centroids then guide the update of the resident latent prototypes $p_k^z$. Specifically, each computed cluster centroid is utilized to update the closest latent prototype $p_k^z$ via an Exponential Moving Average (EMA), which ensures a smooth evolution of these resident prototypes.

Then our primary pathway follows the refinement module that we decode the refined latent prototypes back to the input space, $\hat{p}_k^x = D_\psi(p_k^z)$, and use $\hat{p}_k^x$ to update the active time-domain prototypes. In addition, we incorporate an input-space anchoring (ISA) signal as an auxiliary stabilizer, which is more relied upon early in training and gradually down-weighted as the latent prototypes become reliable.

- **Input-Space Anchoring:** This strategy provides initial, data-driven estimates for time-domain patterns. Raw input segments $S_{\text{raw}}$ are clustered in the time domain. The resulting cluster centroids, denoted as $\{c_k\}$, serve as direct, empirically derived reference points.
- **Latent-to-Time-Domain Decoding Consistency:** This ensures refined latent prototypes $p_k^z$ map to meaningful

time-domain patterns. A decoder $D_\psi$ generates time series $\hat{p}_k^x = D_\psi(p_k^z)$ from each latent prototype.

The final active time-domain prototypes $p_k^x$ are then updated by fusing information from both sources: a weighted combination of the empirically derived centroids $\{c_k\}$ from ISA and the decoder outputs $\{\hat{p}_k^x\}$. This comprehensive and adaptive strategy aims to refine both latent and time-domain prototypes to robustly capture true semantics, supporting the iterative improvement principles.

**3.2.3. Learning Objectives**  The encoder $f_\theta$ learns representations $z_i = f_\theta(x_i)$ and $\tilde{z}_i = f_\theta(\tilde{x}_i)$. The learning process is driven by a two-component contrastive loss function. The first component is an intra-instance temporal contrastive loss ($L_{intra}$), inspired by the local contrast mechanism (Tonekaboni et al., 2021), designed to model fine-grained temporal patterns. The second component is an inter-instance semantic contrastive loss ($L_{inter}$). This loss learns robust semantic relationships by contrasting instances based on their association with learned prototypes, effectively aligning instances to these semantic anchors. The detailed expressions for these losses are provided in Appendix A. The overall learning objective is set as a weighted sum of these components:

$$\mathcal{L}_{total} = \lambda_{intra} L_{intra} + \lambda_{inter} L_{inter}. \tag{5}$$

# 4. Empirical Evaluation

To thoroughly evaluate the performance of ProSAR, we conduct extensive experiments on both the time series forecasting and classification tasks.

To further assess the transferability and label efficiency of the learned representations, we additionally evaluate ProSAR under transfer-learning and semi-supervised classification settings. Furthermore, detailed ablation studies are performed to analyze the contribution of each component within the ProSAR framework. Implementation details are provided in Appendix B, while additional quantitative results and visual analyses are provided in Appendices C and D.

## 4.1. Evaluation Setup

**Forecasting.**  Our forecasting evaluation uses five widely adopted benchmarks: ETTh1, ETTh2, ETTm1, Electricity, and Weather (Zhou et al., 2021). We compare ProSAR with representative self-supervised time-series methods, including TNC (Tonekaboni et al., 2021), TS-TCC (Eldele et al., 2021), TS2Vec (Yue et al., 2022), CoST (Woo et al., 2022), InfoTS (Luo et al., 2023), AutoTCL (Zheng et al., 2024), TimesURL (Liu & Chen, 2024), PPT (Kim et al., 2025), and FreRA (Tian et al., 2025). Following the linear evaluation

protocol of TS2Vec (Yue et al., 2022), we freeze the pretrained encoder and train a linear predictor for downstream forecasting. ProSAR adopts the CoST-style dilated CNN encoder (Woo et al., 2022), and performance is evaluated by MSE and MAE under both univariate and multivariate settings.

**Classification.**  For standard time-series classification, ProSAR is evaluated on the UEA multivariate time series archive (Dau et al., 2019). We compare with TNC (Tonekaboni et al., 2021), TS-TCC (Eldele et al., 2021), TS2Vec (Yue et al., 2022), InfoTS (Luo et al., 2023), AutoTCL (Zheng et al., 2024), TimesURL (Liu & Chen, 2024), PPT (Kim et al., 2025), and FreRA (Tian et al., 2025). Following common practice in time-series representation learning, the pretrained encoder extracts instance-level representations, and an RBF-kernel SVM is trained on the training split and evaluated on the test split. We report classification accuracy (ACC) and average rank (RANK) across datasets.

**Transfer Learning.**  Following FreRA (Tian et al., 2025), we evaluate cross-domain transfer on SHAR, where subject-wise domains introduce distribution shifts. We compare with FreRA (Tian et al., 2025), InfoTS (Luo et al., 2023), AutoTCL (Zheng et al., 2024), TS2Vec (Yue et al., 2022), TNC (Tonekaboni et al., 2021), TS-TCC (Eldele et al., 2021), and SoftCLT (Lee et al., 2024), with baseline results taken from FreRA for fair comparison. The encoder is pretrained on source domains and evaluated on an unseen target domain. Accuracy is used as the evaluation metric.

**Semi-Supervised Classification.**  Following Soft-CLT (Lee et al., 2024), we evaluate the 1% labeled-data setting on eight classification datasets: HAR, Epilepsy, Wafer, FordA, FordB, POC, StarLightCurves, and ElectricDevices. We compare with FreRA (Tian et al., 2025), AutoTCL (Zheng et al., 2024), TS2Vec (Yue et al., 2022), SoftCLT (Lee et al., 2024), TS-TCC (Eldele et al., 2021), and the TS-TCC-based variant of SoftCLT. The encoder is first pretrained on unlabeled data and then fine-tuned with only 1% labeled samples. We report classification accuracy (ACC) and macro-F1 score (MF1).

## 4.2. Quantitative Results

**Forecasting.**  The average forecasting performance across all the datasets for univariate and multivariate settings is presented in Table 2 and Table 3, where bold and underlined values represent the best and the second-best results, respectively. ProSAR consistently surpasses AutoTCL, indicating that the prototype-guided semantic augmentation performs better than the other augmentation schemes. In the univariate setting, ProSAR reduces the average MSE/MAE from

*Table 2.* Univariate time series forecasting results.

| Dataset | ProSAR | | AutoTCL | | FreRA | | PPT | | TimesURL | | InfoTS | | TS2Vec | | TNC | | TS–TCC | | CoST | |
|---|---|---|---|---|---|---|---|---|---|---|---|---|---|---|---|---|---|---|---|---|
| | MSE | MAE | MSE | MAE | MSE | MAE | MSE | MAE | MSE | MAE | MSE | MAE | MSE | MAE | MSE | MAE | MSE | MAE | MSE | MAE |
| ETTh$_1$ | **0.068** | **0.198** | 0.076 | 0.207 | 0.079 | 0.213 | 0.087 | 0.221 | 0.090 | 0.219 | 0.091 | 0.227 | 0.110 | 0.252 | 0.150 | 0.303 | 0.168 | 0.316 | 0.091 | 0.228 |
| ETTh$_2$ | **0.142** | **0.289** | 0.158 | 0.299 | 0.171 | 0.315 | 0.159 | 0.303 | 0.151 | 0.295 | 0.149 | 0.299 | 0.170 | 0.321 | 0.168 | 0.322 | 0.298 | 0.428 | 0.161 | 0.307 |
| ETTm$_1$ | **0.045** | **0.151** | 0.046 | 0.154 | 0.051 | 0.162 | 0.062 | 0.174 | 0.053 | 0.175 | 0.050 | 0.157 | 0.069 | 0.186 | 0.069 | 0.191 | 0.158 | 0.299 | 0.054 | 0.164 |
| Elec | **0.338** | **0.328** | 0.366 | 0.345 | 0.360 | 0.339 | 0.389 | 0.369 | 0.374 | 0.356 | 0.368 | 0.348 | 0.393 | 0.370 | 0.378 | 0.359 | 0.511 | 0.603 | 0.375 | 0.353 |
| WTH | **0.160** | **0.285** | 0.160 | 0.287 | 0.169 | 0.301 | 0.177 | 0.307 | 0.177 | 0.302 | 0.176 | 0.304 | 0.181 | 0.308 | 0.175 | 0.303 | 0.302 | 0.442 | 0.183 | 0.307 |
| Avg. | **0.151** | **0.250** | 0.161 | 0.258 | 0.166 | 0.266 | 0.175 | 0.275 | 0.169 | 0.269 | 0.167 | 0.267 | 0.185 | 0.287 | 0.188 | 0.296 | 0.287 | 0.418 | 0.173 | 0.272 |

*Table 3.* Multivariate time series forecasting results.

| Dataset | ProSAR | | AutoTCL | | FreRA | | PPT | | TimesURL | | InfoTS | | TS2Vec | | TNC | | TS-TCC | | CoST | |
|---|---|---|---|---|---|---|---|---|---|---|---|---|---|---|---|---|---|---|---|---|
| | MSE | MAE | MSE | MAE | MSE | MAE | MSE | MAE | MSE | MAE | MSE | MAE | MSE | MAE | MSE | MAE | MSE | MAE | MSE | MAE |
| ETTh$_1$ | **0.625** | **0.566** | 0.656 | 0.590 | 0.646 | 0.584 | 0.750 | 0.650 | 0.731 | 0.645 | 0.784 | 1.622 | 0.788 | 0.646 | 0.904 | 0.702 | 0.748 | 0.635 | 0.650 | 0.585 |
| ETTh$_2$ | 1.213 | 0.819 | **1.191** | **0.815** | 1.397 | 0.893 | 1.529 | 0.928 | 1.514 | 0.926 | 1.474 | 0.914 | 1.566 | 0.937 | 1.869 | 1.053 | 2.120 | 1.109 | 1.283 | 0.851 |
| ETTm$_1$ | **0.396** | **0.434** | 0.409 | 0.441 | 0.445 | 0.467 | 0.562 | 0.580 | 0.561 | 0.584 | 0.568 | 0.521 | 0.628 | 0.553 | 0.740 | 0.599 | 0.612 | 0.564 | 0.409 | 0.439 |
| Elec | **0.159** | **0.264** | 0.175 | 0.272 | 0.182 | 0.278 | 0.213 | 0.312 | 0.202 | 0.299 | 0.289 | 0.376 | 0.319 | 0.397 | 0.387 | 0.446 | 0.511 | 0.602 | 0.165 | 0.268 |
| WTH | **0.412** | **0.451** | 0.423 | 0.457 | 0.429 | 0.462 | 0.445 | 0.466 | 0.447 | 0.469 | 0.455 | 0.472 | 0.451 | 0.474 | 0.441 | 0.466 | 0.483 | 0.535 | 0.430 | 0.464 |
| Avg | **0.561** | **0.507** | 0.571 | 0.515 | 0.620 | 0.537 | 0.700 | 0.587 | 0.691 | 0.585 | 0.714 | 0.781 | 0.750 | 0.601 | 0.868 | 0.653 | 0.895 | 0.689 | 0.587 | 0.521 |

*Table 4.* Classification result of the UEA dataset.

| Metric | ProSAR | AutoTCL | FreRA | PPT | TimesURL | InfoTS | TS2Vec | TNC | TS–TCC |
|---|---|---|---|---|---|---|---|---|---|
| Avg. ACC | **0.764** | 0.742 | 0.754 | 0.735 | 0.752 | 0.730 | 0.704 | 0.670 | 0.668 |
| Avg. RANK | **1.867** | 3.067 | 2.900 | 3.200 | 2.233 | 3.133 | 4.367 | 5.500 | 5.367 |

0.161/0.258 of AutoTCL to 0.151/0.250, corresponding to a 6.2% lower MSE and a 3.5% lower MAE. In the multivariate setting, ProSAR also improves the average MSE/MAE from 0.571/0.515 of AutoTCL to 0.561/0.507, achieving a 1.8% MSE reduction and a 1.6% MAE reduction. It further wins on 4 of 5 datasets when averaging across prediction lengths per dataset, reflecting stronger representations that preserve key dynamics while ensuring view diversity.

**Classification.** Classification results across the 30 UEA datasets is provided in Table 4, where bold and underlined values represent the best and the second-best results, respectively. ProSAR attains the highest mean accuracy of 0.764 and the best mean rank of 1.867 among all methods. Compared with FreRA, ProSAR improves the mean accuracy from 0.754 to 0.764. Compared with TimesURL, ProSAR reduces the average rank from 2.233 to 1.867, yielding an improvement of 0.366. These results indicate that ProSAR learns more discriminative representations for time-series classification.

**Transfer Learning.** Table 5 reports the SHAR transfer results. ProSAR achieves the best performance in 7 out of 8 settings and ranks second in the remaining one, improving the average accuracy from 0.596 of FreRA to 0.615. The advantage is more evident under the data-scarce transfer setting with 3 source domains, where ProSAR improves the average accuracy from 0.525 to 0.553 and ranks first on all four target domains. For TD=2, ProSAR improves over FreRA from 0.467 to 0.516, and for TD=5, it improves from 0.366 to 0.406. These results show that the learned representations transfer effectively across subject domains.

**Semi-Supervised Classification.** Table 6 reports the 1% labeled-data results. ProSAR obtains the best ACC on 7 out of 8 datasets and achieves the highest average ACC/MF1 of 0.853/0.833. Compared with FreRA, ProSAR improves the average ACC/MF1 from 0.836/0.805 to 0.853/0.833, and it consistently outperforms AutoTCL on all datasets. The gains are clear on POC, where ProSAR achieves 0.720/0.671 compared with 0.682/0.633 of FreRA, and on ElectricDevices, where ProSAR achieves 0.651/0.632 compared with 0.633/0.553 of FreRA. These results demonstrate improved label efficiency under low-label SSL scenarios.

### 4.3. Ablation Studies and Computational Efficiency

To evaluate ProSAR's core contributions, we compare the full model against variants disabling specific mechanisms: DTW-based segmentation (w/o DTW-Seg), frequency alignment (w/o STFT), input-space grounding (w/o Dual-Proto), dynamic clustering (w/o Clustering), and decoder feedback (w/o Decoding). As summarized in Table 11 in Appendix C.4, all the variants degrade from the full model (MSE=0.151), confirming that each component is contributive. The significant drop in w/o DTW-Seg (0.172) and w/o STFT (0.178) empirically validates that explicit semantic segmentation and phase alignment are superior to uniform processing for capturing temporal dynamics. Regarding refinement, w/o Clustering suffers a severe penalty (0.175), while w/o Decoding (0.164) and w/o Dual-Proto (0.158) show a tangible degradation, underscoring that anchoring latent concepts to time-domain centroids via a closed feedback loop provides essential structural constraints. Collectively, these results distinguish ProSAR from the existing baselines, proving that active prototype feedback offers a

*Table 5.* Transfer-learning classification accuracy on SHAR. No. SD denotes the number of source domains and TD denotes the target domain. The best results are in **bold**, and the second-best results are underlined.

| No. SD | TD | ProSAR | FreRA | InfoTS | AutoTCL | TS2Vec | TNC | TS-TCC | SoftCLT |
|---|---|---|---|---|---|---|---|---|---|
| 3 | 1 | **0.604** | 0.602 | 0.367 | 0.464 | 0.430 | 0.133 | 0.495 | 0.505 |
| 3 | 2 | **0.516** | 0.467 | 0.369 | 0.278 | 0.317 | 0.145 | 0.410 | 0.407 |
| 3 | 3 | **0.684** | 0.665 | 0.516 | 0.414 | 0.523 | 0.217 | 0.464 | 0.530 |
| 3 | 5 | **0.406** | 0.366 | 0.081 | 0.245 | 0.050 | 0.143 | 0.362 | 0.339 |
| 19 | 1 | **0.654** | 0.628 | 0.599 | 0.497 | 0.568 | 0.117 | 0.578 | 0.581 |
| 19 | 2 | **0.664** | 0.652 | 0.455 | 0.372 | 0.640 | 0.148 | 0.647 | 0.581 |
| 19 | 3 | **0.697** | 0.691 | 0.563 | 0.408 | 0.502 | 0.135 | 0.592 | 0.559 |
| 19 | 5 | 0.695 | **0.698** | 0.638 | 0.430 | 0.658 | 0.204 | 0.612 | 0.567 |

*Table 6.* Semi-supervised classification results with 1% labeled data. Results are reported as ACC/MF1, with the best values in **bold**.

| Dataset | ProSAR | FreRA | AutoTCL | TS2Vec | SoftCLT (TS2Vec-based) | TS-TCC | SoftCLT (TS-TCC-based) |
|---|---|---|---|---|---|---|---|
| HAR | **0.915 / 0.915** | 0.901 / 0.903 | 0.783 / 0.774 | 0.886 / 0.885 | 0.910 / 0.910 | 0.705 / 0.695 | 0.829 / 0.828 |
| Epilepsy | **0.964 / 0.963** | 0.957 / 0.955 | 0.928 / 0.924 | 0.958 / 0.934 | 0.963 / 0.941 | 0.912 / 0.892 | 0.956 / 0.956 |
| Wafer | **0.978** / 0.941 | 0.964 / 0.900 | 0.936 / 0.784 | 0.679 / 0.561 | 0.953 / 0.881 | 0.932 / 0.767 | 0.965 / **0.965** |
| FordA | **0.896 / 0.895** | 0.872 / 0.872 | 0.853 / 0.853 | 0.864 / 0.864 | 0.871 / 0.871 | 0.806 / 0.800 | 0.815 / 0.812 |
| FordB | 0.782 / 0.782 | 0.774 / 0.774 | 0.751 / 0.751 | 0.654 / 0.654 | 0.679 / 0.679 | **0.786 / 0.786** | 0.748 / 0.748 |
| POC | **0.720 / 0.671** | 0.682 / 0.633 | 0.675 / 0.626 | 0.631 / 0.628 | 0.636 / 0.628 | 0.638 / 0.481 | 0.654 / 0.646 |
| StarLightCurves | **0.914 / 0.863** | 0.908 / 0.853 | 0.893 / 0.804 | 0.829 / 0.606 | 0.856 / 0.629 | 0.860 / 0.792 | 0.860 / 0.793 |
| ElectricDevices | **0.651 / 0.632** | 0.633 / 0.553 | 0.629 / 0.551 | 0.576 / 0.486 | 0.620 / 0.530 | 0.636 / 0.564 | 0.646 / **0.632** |

superior stability compared to purely implicit augmentation strategies.

A potential concern with DTW utilized in our ProSAR is the computational complexity. However, we empirically verify that our prototype-guided alignment introduces negligible overhead. Powered by a CUDA-optimized implementation, the alignment step requires only 1 ms per iteration, resulting in a marginal 2.5% increase in total training time. Detailed runtime and scalability analyses, including data-size scaling, sequence-length scaling, and module-level profiling, are provided in Appendix C.9.

### 4.4. Additional Results in the Appendix

Beyond the main results, Appendix C reports detailed forecasting and classification results, ablation studies, sensitivity analyses, and computational efficiency analyses. Appendix D further provides qualitative visualizations of the learned prototypes, alignment mechanisms, and augmentation process.

### 4.5. Limitation

While ProSAR demonstrates performance, it relies on hyperparameters that may require tuning for specific datasets. Additionally, though prototypes serve as explicit anchors, how to ensure a consistent capturing of human-interpretable semantics across complex domains remains non-trivial. Finally, a further extension of ProSAR to incorporate some large-scale foundation models represents another promising direction for future works.

## 5. Conclusion

In this work, we addressed the interpretability and semantic preservation challenges in time series augmentation by proposing ProSAR. Unlike prior methods that extract semantic parts implicitly, ProSAR leveraged the learnable prototypes as explicit anchors to guide semantic segmentation and proposed a new augmentation method within an information-theoretic framework. By establishing a closed feedback loop between augmentation, latent representation learning and time-domain prototype refinement, our approach has enabled the augmentation strategies to co-evolve with the model's semantic understanding.

## Acknowledgements

This work was supported in part by the National Natural Science Foundation of China under Grants 62125109, 62120106007, U24A20251, 62431017, 62320106003 and 62371288, in part by the Fundamental and Interdisciplinary Disciplines Breakthrough Plan of the Ministry of Education of China under Grant JYB2025XDXM611, in part by the National Key Research and Development Program of China under Grants 2025YFF0515602 and 2025YFF0515604, and in part by the AI for Science Program, Shanghai Municipal Commission of Economy and Informatization under Grant 2025-GZL-RGZN-BTBX-02022.

## Impact Statement

This paper aims to advance machine learning by improving self-supervised representation learning for multivariate time series via prototype-guided semantic augmentation. Bet-

ter time-series representations may benefit the downstream applications, such as healthcare monitoring, industrial IoT maintenance, and other decision-support systems that rely on temporal data. Our work does not aim to introduce new data collection procedures, while we encourage researchers and engineers to follow the established privacy-preserving protocols, conduct bias/robustness evaluations under distribution shifts, and apply domain-appropriate safeguards when deploying these learned representations in the real-world systems.

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

# Appendix

This appendix provides supplementary material to the main paper, organized into the following five main sections. Section A offers a comprehensive description of our ProSAR algorithm, detailing the overall training framework and the specific formulations of the intra-instance and inter-instance contrastive loss functions. Section B outlines the experimental setup, covering baseline implementations, ProSAR's hyperparameter configurations, evaluation protocols for forecasting, classification, transfer learning, and semi-supervised classification, and the computational environment. Section C presents extensive additional experimental results. This includes detailed performance tables for all forecasting and classification benchmarks, results from ablation studies, discussion on network architectures, parameter sensitivity analysis, and model convergence plots. Finally, Section D provides qualitative insights through various visualizations, illustrating learned prototypes, alignment mechanisms, and the step-by-step augmentation process. Detailed theoretical derivations and proofs (Section E) are also provided at the end of this appendix.

## A. Algorithm Details

### A.1. Overall ProSAR Algorithm

The ProSAR framework operates through an iterative process that synergistically refines prototypes and guides semantic augmentation. The overall training loop is detailed in Algorithm 1.

---

**Algorithm 1** ProSAR training algorithm

---

**Require:** Raw time series dataset $\mathcal{D}$; encoder $f_\theta$; decoder $D_\psi$; number of prototypes $K$; number of time-domain prototypes $K_t$; learning rate $\eta_\theta$; batch size $B$; number of epochs $E_{\max}$.
 1: Initialize encoder $f_\theta$ and decoder $D_\psi$.
 2: Initialize latent prototypes $\{p_k^z\}_{k=1}^K$ and time-domain prototypes $\{p_j^x\}_{j=1}^{K_t}$.
 3: **for** epoch $= 1$ to $E_{\max}$ **do**
 4:    **for** each batch $X_B \subset \mathcal{D}$ **do**
 5:       **Step 1: Prototype-guided semantic view generation.**
 6:       For each $x \in X_B$, identify semantic segments $\{S_{\text{raw},j}\}$ using current $\{p_j^x\}$ and DTW alignment.
 7:       Define $x_S = x \odot M_x$ and $x_N = x \odot (1 - M_x)$.
 8:       Transform $x_S \to x_S'$ and perturb $x_N \to x_N'$.
 9:       Assemble $\tilde{x} = (x_S' \odot M_x) + (x_N' \odot (1 - M_x))$ and obtain $\tilde{X}_B$.
10:       **Step 2: Encoder update and contrastive loss.**
11:       Obtain $Z_B = f_\theta(X_B)$ and $\tilde{Z}_B = f_\theta(\tilde{X}_B)$.
12:       Calculate $L_{\text{intra}}$ and $L_{\text{inter}}$.
13:       $\mathcal{L}_{\text{total}} = \lambda_{\text{intra}} L_{\text{intra}} + \lambda_{\text{inter}} L_{\text{inter}}$.
14:       Update $f_\theta$ by minimizing $\mathcal{L}_{\text{total}}$.
15:       **Step 3: Prototype refinement.**
16:       Cluster $Z_B \cup \tilde{Z}_B$ using FINCH to obtain centroids $\{\mu_c\}$.
17:       Update latent prototypes $\{p_k^z\}$ via EMA.
18:       Extract raw segments $S_{\text{raw}}$ and cluster them to obtain $\{c_j^{\text{isg}}\}$.
19:       Generate decoded candidates $\hat{p}_k^x = D_\psi(p_k^z)$.
20:       Update $\{p_j^x\}$ by fusing $\{c_j^{\text{isg}}\}$ and relevant $\{\hat{p}_k^x\}$.
21:    **end for**
22: **end for**
23: **return** trained encoder $f_\theta$ and refined prototypes $\{p_k^z\}, \{p_j^x\}$.

---

### A.2. Details of Prototype Feedback and Refinement

This subsection provides additional details on the decoder-based feedback, the fusion of time-domain prototype candidates, and the prototype-number selection strategy.

**Decoder warmup and feedback mechanism.** The decoder $D_\psi$ is used to provide latent-to-time-domain feedback for refining the active time-domain prototypes. To make this feedback meaningful, we adopt a two-stage training strategy. In the first stage, we perform a short reconstruction warmup to initialize the encoder-decoder mapping. Given an input time series $x_i$, the encoder maps it to the latent representation $z_i = f_\theta(x_i)$, and the decoder reconstructs the input as $\hat{x}_i = D_\psi(z_i)$. The warmup objective is the mean squared reconstruction loss:

$$\mathcal{L}_{\text{rec}} = \frac{1}{B} \sum_{i=1}^{B} \|D_\psi(f_\theta(x_i)) - x_i\|_2^2, \tag{6}$$

where $B$ denotes the batch size. This warmup stage establishes an initial mapping from latent representations back to time-domain patterns, which is required for decoding latent prototypes into time-domain prototype candidates.

After the warmup stage, the decoder is fixed during the main contrastive training stage. The encoder $f_\theta$ is optimized by the contrastive objective, while the latent prototypes are updated by clustering the learned representations. Although $D_\psi$ is frozen, the decoded feedback remains dynamic because the latent prototypes $p_k^{\text{z}}$ are continuously refreshed during training. Specifically, each updated latent prototype is decoded as

$$\hat{p}_k^{\text{x}} = D_\psi(p_k^{\text{z}}), \tag{7}$$

and the decoded time-domain candidate $\hat{p}_k^{\text{x}}$ is then used to refine the active time-domain prototype $p_k^{\text{x}}$. Therefore, the decoder does not introduce additional optimization objectives in the main training stage, but serves as a fixed feedback bridge that transfers the evolving latent semantics back to the time-domain prototype space.

**Matching, weighted fusion, and EMA update.** The active time-domain prototypes are refined by combining two sources of information: input-space anchoring centroids and decoded latent prototypes. Let $\{c_j\}_{j=1}^{K_t}$ denote the empirical time-domain centroids obtained from input-space anchoring, and let $\hat{p}_k^{\text{x}} = D_\psi(p_k^{\text{z}})$ denote the decoded candidate from the $k$-th latent prototype. For each active prototype, we first establish its correspondence to an input-space centroid by nearest-distance matching:

$$m(k) = \arg \min_{j \in \{1, \dots, K_t\}} d\left(\hat{p}_k^{\text{x}}, c_j\right), \tag{8}$$

where $d(\cdot, \cdot)$ denotes the distance used for comparing time-domain prototype sequences. In practice, this matching step provides a one-to-one update correspondence for each active prototype candidate.

The matched input-space centroid and the decoded latent prototype are then fused as

$$\bar{p}_k^{\text{x}} = \alpha_t c_{m(k)} + (1 - \alpha_t)\hat{p}_k^{\text{x}}, \tag{9}$$

where $\alpha_t \in [0, 1]$ controls the relative contribution of input-space anchoring and latent-to-time feedback at training epoch $t$. At the early stage, a larger $\alpha_t$ is used to rely more on empirical time-domain centroids, which stabilizes the prototype initialization. As training proceeds, $\alpha_t$ is gradually decayed so that the refined latent semantics can contribute more to the time-domain prototypes.

Finally, the active time-domain prototype is updated by exponential moving average:

$$p_k^{\text{x}} \leftarrow \rho p_k^{\text{x}} + (1 - \rho)\bar{p}_k^{\text{x}}, \tag{10}$$

where $\rho \in [0, 1]$ is the EMA coefficient. This EMA update avoids abrupt prototype changes and makes the prototype-refinement process stable across training iterations.

**Prototype-number selection and close-prototype handling.** The numbers of latent prototypes and time-domain prototypes are treated as dataset-dependent hyperparameters and selected using the validation set. In practice, ProSAR is not sensitive to a highly specific prototype number. On the univariate forecasting benchmark, increasing the prototype number from 16 to 32 improves the average MSE/MAE from 0.153/0.252 to 0.151/0.250, while further increasing it to 64 keeps the performance almost unchanged at 0.151/0.250. This indicates that a moderate number of prototypes is sufficient to capture the major semantic patterns, and further increasing the prototype number brings limited additional benefit.

When multiple prototypes become close to each other, ProSAR does not perform hard replacement or direct deletion. Instead, the matching and EMA smoothing mechanism updates each active prototype gradually based on both the decoded latent

feedback and the input-space anchoring signal. Thus, transient similarity between prototypes does not immediately cause instability. If two prototypes are partially redundant, their future updates can still diverge when they are matched to different input-space patterns or latent clusters in subsequent iterations. This design makes the refinement process robust to temporary prototype redundancy and initialization noise.

### A.3. Loss Function Details

The overall learning objective is $\mathcal{L}_{\text{total}} = \lambda_{\text{intra}} L_{\text{intra}} + \lambda_{\text{inter}} L_{\text{inter}}$.

#### A.3.1. INTRA-INSTANCE TEMPORAL CONTRASTIVE LOSS ($L_{\text{INTRA}}$)

The intra-instance temporal contrastive loss ($L_{\text{intra}}$) is designed to model fine-grained temporal patterns by enforcing consistency between representations of nearby timestamps within the same augmented instance, while distinguishing them from representations of distant timestamps. This is inspired by the local contrast mechanism (Tonekaboni et al., 2021) and similar to approaches like TNC (Tonekaboni et al., 2021) or the local module in TS2Vec (Yue et al., 2022).

Given an augmented view $\tilde{x}$ and its latent representation sequence $\tilde{z} = (\tilde{z}_1, \tilde{z}_2, \ldots, \tilde{z}_{T'}')$, where $T'$ is the length of the latent sequence. For each anchor timestamp $t_a$, we sample a positive timestamp $t_p$ from its temporal neighborhood (e.g., within a window $W_{intra}$) and a set of negative timestamps $\{t_{n,j}\}$ from outside this neighborhood. The loss for a single anchor $\tilde{z}_{t_a}$ in an instance can be formulated using InfoNCE:

$$L_{\text{intra}}(\tilde{z}_{t_a}) = -\log \frac{\exp(\text{sim}(\tilde{z}_{t_a}, \tilde{z}_{t_p})/\tau_{intra})}{\exp(\text{sim}(\tilde{z}_{t_a}, \tilde{z}_{t_p})/\tau_{intra}) + \sum_j \exp(\text{sim}(\tilde{z}_{t_a}, \tilde{z}_{t_{n,j}})/\tau_{intra})}$$

where $\text{sim}(\cdot, \cdot)$ is a similarity function (e.g., cosine similarity) and $\tau_{intra}$ is a temperature hyperparameter. The total $L_{\text{intra}}$ is averaged over all anchor timestamps and all instances in the batch. This loss encourages the model to learn representations that are smooth over short temporal ranges yet discriminative over longer ranges.

#### A.3.2. INTER-INSTANCE SEMANTIC CONTRASTIVE LOSS ($L_{\text{INTER}}$)

The inter-instance semantic contrastive loss ($L_{\text{inter}}$) leverages learned latent prototypes $\{p_k^z\}$ to learn robust semantic relationships. It comprises two components: an inter-instance term ($L_{\text{inter\_inst}}$) and an instance-to-prototype term ($L_{\text{inter\_proto}}$), such that $L_{\text{inter}} = L_{\text{inter\_inst}} + L_{\text{inter\_proto}}$. These components are inspired by principles from PCL (Li et al., 2021a) and MHCCL (Meng et al., 2023).

For a batch of instances $X_B$ with representations $Z_B = f_\theta(X_B)$ (and augmentations $\tilde{Z}_B = f_\theta(\tilde{X}_B)$), each $z_i$ (or $\tilde{z}_i$) is assigned to its closest prototype $p_{c(i)}^l$.

**Inter-instance Contrastive Loss ($L_{\text{inter\_inst}}$)** This term promotes similarity for an anchor $z_i$ with its augmentation $\tilde{z}_i$ and other instances/augmentations $z_j, \tilde{z}_j$ assigned to the same prototype $p_{c(i)}^l$. It distinguishes them from instances/augmentations $z_k, \tilde{z}_k$ of different prototypes. The positive set for an anchor $z_i$ is $\text{Pos}_{\text{inst}}(z_i) = \{\tilde{z}_i\} \cup \{z_j, \tilde{z}_j \mid c(j) = c(i), j \neq i \text{ and } z_j, \tilde{z}_j \text{ from batch}\}$. The negative set $\text{Neg}_{\text{inst}}(z_i) = \{z_k, \tilde{z}_k \mid c(k) \neq c(i) \text{ and } z_k, \tilde{z}_k \text{ from batch}\}$. The InfoNCE loss for $z_i$ is:

$$L_{\text{inter\_inst}}(z_i) = -\log \frac{\sum_{e_p \in \text{Pos}_{\text{inst}}(z_i)} \exp(\text{sim}(z_i, e_p)/\tau_{\text{inst}})}{\sum_{e_p \in \text{Pos}_{\text{inst}}(z_i)} \exp(\text{sim}(z_i, e_p)/\tau_{\text{inst}}) + \sum_{e_n \in \text{Neg}_{\text{inst}}(z_i)} \exp(\text{sim}(z_i, e_n)/\tau_{\text{inst}})}$$

where $\tau_{\text{inst}}$ is a temperature. $L_{\text{inter\_inst}}$ is the batch-averaged loss, considering both $z_i$ and $\tilde{z}_i$ as anchors.

**Instance-to-Prototype Contrastive Loss ($L_{\text{inter\_proto}}$)** This term aligns an anchor representation $z_i$ with its assigned prototype $p_{c(i)}^l$ (positive) and separates it from all other prototypes $p_k^z, k \neq c(i)$ (negatives). The loss for an anchor $z_i$ is:

$$L_{\text{inter\_proto}}(z_i) = -\log \frac{\exp(\text{sim}(z_i, p_{c(i)}^l)/\tau_{\text{proto}})}{\sum_{k=1}^{K} \exp(\text{sim}(z_i, p_k^z)/\tau_{\text{proto}})}$$

where $K$ is the number of prototypes and $\tau_{\text{proto}}$ is a temperature. $L_{\text{inter\_proto}}$ is the batch-averaged loss, considering both $z_i$ and $\tilde{z}_i$ as anchors.

Thus, the total $L_{\text{inter}}$ encourages both intra-prototype instance cohesion and precise instance-prototype alignment.

# B. Experimental Settings

## B.1. Baseline Implementation Details

For most baselines, we adopt results reported in their original papers or established benchmark papers if experimental setups are identical. The linear evaluation protocol for contrastive methods in forecasting follows Yue et al. (2022).

## B.2. Hyperparameter Settings for ProSAR

Key hyperparameters for ProSAR are carefully tuned to achieve optimal performance. Table 7 summarizes common settings or search ranges for these crucial parameters. This includes aspects such as the general training setup (like batch size and optimizer); configurations for the encoder and decoder networks; parameters for ProSAR's prototype system; details of the semantic augmentation process; and settings for the contrastive loss functions. Specific values used for each reported result will be available. As noted in the main paper, ProSAR's encoder for forecasting is based on a modified CoST backbone (Woo et al., 2022) (with its seasonal disentangler module omitted to isolate ProSAR's contributions), while the TS2Vec (Yue et al., 2022) encoder architecture is adopted for classification tasks.

*Table 7.* Hyperparameter ranges/settings for ProSAR.

| Hyperparameter | Value Range / Setting | Notes |
|---|---|---|
| **General Setup** | | |
| Batch Size | {32, 64, 128, 256} | Depends on memory constraints |
| Epochs | 50 - 200 | Early stopping based on validation performance |
| Optimizer | AdamW | Betas (0.9, 0.999), Weight decay $10^{-4}$ |
| **Encoder Architecture** ($f_\theta$) | | |
| Encoder Learning Rate | {1e-5, 5e-5, 1e-4, 5e-4} | For $f_\theta$ |
| Backbone Type | CoST/ TS2Vec | As per main paper |
| Dropout Rate | {0.1, 0.2, 0.3, 0.5} | Within encoder backbone |
| **Decoder Architecture** ($D_\psi$) | | |
| Architecture Type | MLP | As described in main paper |
| Layers | 3 | Implementation specific |
| Hidden Dimension | 128 | Implementation specific |
| **Prototype System** | | |
| Number of Prototypes ($K$) | 32 | Dataset dependent, tune via validation |
| Number of Time-Domain Prototypes ($K_t$) | 32 | Dataset dependent, tune via validation |
| Latent Proto. EMA Momentum ($\alpha_z$) | 0.99 | For $p_k^z$ update; typically high for stability |
| Time Proto. Update Fusion Weight ($\alpha_x$) | 0.5 | For combining ISG and decoded $p_k^x$ |
| **Augmentation Parameters** | | |
| Frequency Noise Level ($x_S$) | 0.1 | For semantic part transform |
| Non-Semantic Part Noise Level ($x_N$) | 0.1 | For perturbation of $x_N$ |
| Non-Semantic Part Masking Ratio ($x_N$) | 0.3 | For sub-segment masking in $x_N$ |
| **Contrastive Loss** | | |
| $L_{intra}$ Temp. $\tau_{intra}$ | 0.1 | For intra-instance loss |
| $L_{inter}$ Temp. $\tau_{inter}$ | 0.1 | For inter-instance loss |
| $\lambda_{intra}$ (Loss weight) | {0.1, ..., 1.0} | Weight for $L_{intra}$ |
| $\lambda_{inter}$ (Loss weight) | {0.1, ..., 1.0} | Weight for $L_{inter}$ |

## B.3. Evaluation Protocol Details

For forecasting tasks, representations learned by the pretrained ProSAR encoder are frozen. A linear model with L2 regularization is then trained to map these representations to the future $L_y$ time steps. This linear evaluation protocol is applied uniformly across all contrastive learning baselines for fair comparison. Performance is measured using Mean Squared Error (MSE) and Mean Absolute Error (MAE). For classification tasks, following established protocols (Yue et al., 2022), a Radial Basis Function (RBF) kernel Support Vector Machine (SVM) classifier is trained on the representations of the training set and evaluated on the test set. Performance is reported as classification accuracy (ACC) and average rank

(RANK) across datasets.

### B.4. Compute Resources

All experiments were conducted on a Linux machine equipped with 4 NVIDIA RTX3090 GPUs, each with 24GB of memory. The software environment included CUDA 12.x (inferred from `nvidia-cuda-runtime-cu12 12.6.77` and other `cu12` packages in our environment list) and an NVIDIA Driver Version compatible with CUDA 12.x. We used Python 3.9.21 and PyTorch 2.7.0 to construct our project. Key supporting libraries included NumPy 1.24.4, pandas 2.2.3, and scikit-learn 1.6.1.

### B.5. Evaluation Protocol Details

**Forecasting evaluation.** For forecasting, we follow the standard linear evaluation protocol of TS2Vec (Yue et al., 2022). After self-supervised pretraining, the encoder is frozen and only a linear predictor with $\ell_2$ regularization is trained for downstream forecasting. Given a historical input window of length $L_x$, the model predicts the future horizon of length $L_y$. For univariate forecasting, the output dimension is $L_y$, while for multivariate forecasting with $F$ variables, the output dimension is $L_y \times F$. We report mean squared error and mean absolute error over different prediction horizons. For ProSAR, the encoder follows the CoST-style dilated CNN backbone (Woo et al., 2022), while the seasonal-trend disentangling module is omitted to isolate the effect of our prototype-guided augmentation and refinement mechanism.

**Classification evaluation.** For standard time-series classification, we follow the common evaluation protocol used in time-series representation learning. After self-supervised pretraining, the encoder is frozen and used to extract instance-level representations for the training and test samples. A radial basis function kernel support vector machine is then trained on the representations of the training split and evaluated on the test split. We report classification accuracy for each dataset and average rank across all datasets. This protocol is applied consistently to ProSAR and the compared self-supervised baselines.

**Transfer-learning evaluation.** For the transfer-learning setting, we follow the protocol of FreRA (Tian et al., 2025) on the SHAR dataset. Each subject is treated as a domain, and the encoder is pretrained on a set of source domains and directly evaluated on an unseen target domain. We consider two settings with the number of source domains set to 3 and 19, respectively, where TD denotes the target-domain index. The baseline results are taken from the original FreRA paper, and ProSAR is evaluated under the same setting for a fair comparison. Classification accuracy is used as the evaluation metric.

**Semi-supervised evaluation.** For the semi-supervised classification setting, we follow the 1% labeled-data protocol of SoftCLT (Lee et al., 2024). The encoder is first pretrained on unlabeled data and then fine-tuned with only 1% labeled training samples. Following the compared protocol, results of TS2Vec, SoftCLT, TS-TCC, and the TS-TCC-based SoftCLT variant are taken from the original SoftCLT paper, while ProSAR, FreRA, and AutoTCL are evaluated under the same setting. We report classification accuracy and macro-F1 score.

## C. Additional Experimental Results

### C.1. Forecasting: Detailed Results

This section provides the full forecasting performance tables. Table 8 will show detailed univariate results, and Table 9 will show detailed multivariate results, presenting MSE and MAE for all prediction horizons across all datasets and compared methods.

### C.2. Classification: Detailed Results

Table 10 presents the comprehensive classification accuracy results on the 30 UEA datasets for ProSAR and all baseline methods. ProSAR demonstrates a strong average rank and high accuracy across many datasets.

### C.3. Comparison with Representative Prototype-based Baselines

To rigorously evaluate the effectiveness of ProSAR against state-of-the-art prototype-based contrastive learning approaches, we extended our evaluation to include two representative methods: **MHCCL** (Meng et al., 2023) and **AimTS** (Chen et al., 2025).

Table 8. Univariate time series forecasting results.

| Dataset | $L_y$ | ProSAR MSE | ProSAR MAE | AutoTCL MSE | AutoTCL MAE | FreRA MSE | FreRA MAE | PPT MSE | PPT MAE | TimesURL MSE | TimesURL MAE | InfoTS MSE | InfoTS MAE | TS2Vec MSE | TS2Vec MAE | TNC MSE | TNC MAE | TS–TCC MSE | TS–TCC MAE | CoST MSE | CoST MAE |
|---|---|---|---|---|---|---|---|---|---|---|---|---|---|---|---|---|---|---|---|---|---|
| ETTh₁ | 24 | 0.035 | 0.142 | 0.037 | 0.148 | 0.039 | 0.151 | 0.038 | 0.146 | 0.036 | 0.142 | 0.039 | 0.149 | 0.039 | 0.152 | 0.057 | 0.184 | 0.103 | 0.237 | 0.040 | 0.152 |
| | 48 | 0.051 | 0.169 | 0.054 | 0.176 | 0.058 | 0.183 | 0.058 | 0.184 | 0.054 | 0.146 | 0.056 | 0.179 | 0.062 | 0.191 | 0.094 | 0.239 | 0.139 | 0.279 | 0.060 | 0.186 |
| | 168 | 0.074 | 0.205 | 0.078 | 0.210 | 0.081 | 0.221 | 0.092 | 0.238 | 0.096 | 0.233 | 0.100 | 0.239 | 0.134 | 0.282 | 0.171 | 0.329 | 0.253 | 0.408 | 0.097 | 0.236 |
| | 336 | 0.082 | 0.221 | 0.093 | 0.231 | 0.102 | 0.245 | 0.113 | 0.257 | 0.121 | 0.267 | 0.117 | 0.264 | 0.154 | 0.310 | 0.192 | 0.357 | 0.155 | 0.318 | 0.112 | 0.258 |
| | 720 | 0.100 | 0.253 | 0.120 | 0.272 | 0.116 | 0.267 | 0.132 | 0.282 | 0.145 | 0.307 | 0.141 | 0.302 | 0.163 | 0.327 | 0.235 | 0.408 | 0.190 | 0.337 | 0.148 | 0.306 |
| ETTh₂ | 24 | 0.075 | 0.203 | 0.079 | 0.206 | 0.085 | 0.216 | 0.081 | 0.211 | 0.083 | 0.219 | 0.081 | 0.215 | 0.090 | 0.229 | 0.097 | 0.238 | 0.239 | 0.391 | 0.079 | 0.207 |
| | 48 | 0.111 | 0.250 | 0.117 | 0.255 | 0.134 | 0.268 | 0.115 | 0.252 | 0.116 | 0.219 | 0.115 | 0.261 | 0.124 | 0.273 | 0.131 | 0.281 | 0.260 | 0.405 | 0.118 | 0.259 |
| | 168 | 0.166 | 0.313 | 0.176 | 0.319 | 0.205 | 0.357 | 0.178 | 0.325 | 0.175 | 0.332 | 0.171 | 0.327 | 0.208 | 0.360 | 0.197 | 0.354 | 0.291 | 0.420 | 0.189 | 0.339 |
| | 336 | 0.178 | 0.338 | 0.193 | 0.344 | 0.211 | 0.363 | 0.205 | 0.357 | 0.188 | 0.347 | 0.183 | 0.341 | 0.213 | 0.369 | 0.207 | 0.366 | 0.336 | 0.453 | 0.206 | 0.360 |
| | 720 | 0.182 | 0.343 | 0.223 | 0.373 | 0.220 | 0.371 | 0.215 | 0.371 | 0.186 | 0.352 | 0.194 | 0.357 | 0.214 | 0.374 | 0.207 | 0.370 | 0.362 | 0.472 | 0.214 | 0.371 |
| ETTm₁ | 24 | 0.014 | 0.083 | 0.016 | 0.091 | 0.015 | 0.087 | 0.015 | 0.086 | 0.013 | 0.084 | 0.014 | 0.087 | 0.015 | 0.092 | 0.019 | 0.103 | 0.089 | 0.228 | 0.015 | 0.088 |
| | 48 | 0.024 | 0.114 | 0.026 | 0.120 | 0.025 | 0.118 | 0.026 | 0.121 | 0.024 | 0.177 | 0.025 | 0.117 | 0.027 | 0.126 | 0.036 | 0.142 | 0.134 | 0.280 | 0.025 | 0.117 |
| | 96 | 0.035 | 0.141 | 0.036 | 0.145 | 0.039 | 0.149 | 0.045 | 0.163 | 0.037 | 0.145 | 0.036 | 0.142 | 0.044 | 0.161 | 0.054 | 0.178 | 0.159 | 0.305 | 0.038 | 0.147 |
| | 288 | 0.066 | 0.195 | 0.063 | 0.191 | 0.073 | 0.212 | 0.091 | 0.232 | 0.080 | 0.214 | 0.071 | 0.200 | 0.103 | 0.246 | 0.098 | 0.244 | 0.204 | 0.327 | 0.077 | 0.209 |
| | 672 | 0.088 | 0.223 | 0.090 | 0.225 | 0.102 | 0.243 | 0.133 | 0.267 | 0.114 | 0.255 | 0.102 | 0.240 | 0.156 | 0.307 | 0.136 | 0.290 | 0.206 | 0.354 | 0.113 | 0.257 |
| Elec. | 24 | 0.237 | 0.257 | 0.241 | 0.262 | 0.239 | 0.259 | 0.246 | 0.268 | 0.245 | 0.275 | 0.245 | 0.269 | 0.260 | 0.288 | 0.252 | 0.278 | 0.379 | 0.561 | 0.243 | 0.264 |
| | 48 | 0.278 | 0.284 | 0.287 | 0.292 | 0.285 | 0.288 | 0.306 | 0.311 | 0.295 | 0.307 | 0.294 | 0.301 | 0.319 | 0.324 | 0.300 | 0.308 | 0.453 | 0.600 | 0.292 | 0.300 |
| | 168 | 0.383 | 0.352 | 0.394 | 0.365 | 0.393 | 0.368 | 0.428 | 0.386 | 0.408 | 0.379 | 0.402 | 0.367 | 0.427 | 0.394 | 0.412 | 0.384 | 0.575 | 0.616 | 0.405 | 0.375 |
| | 336 | 0.452 | 0.420 | 0.543 | 0.460 | 0.521 | 0.442 | 0.575 | 0.511 | 0.548 | 0.464 | 0.533 | 0.453 | 0.565 | 0.474 | 0.548 | 0.466 | 0.637 | 0.633 | 0.560 | 0.473 |
| WTH | 24 | 0.091 | 0.208 | 0.093 | 0.211 | 0.093 | 0.210 | 0.096 | 0.213 | 0.093 | 0.211 | 0.101 | 0.222 | 0.096 | 0.215 | 0.102 | 0.221 | 0.221 | 0.386 | 0.096 | 0.213 |
| | 48 | 0.131 | 0.256 | 0.131 | 0.256 | 0.137 | 0.261 | 0.141 | 0.262 | 0.131 | 0.255 | 0.141 | 0.266 | 0.139 | 0.264 | 0.139 | 0.264 | 0.255 | 0.406 | 0.138 | 0.262 |
| | 168 | 0.180 | 0.309 | 0.182 | 0.311 | 0.197 | 0.327 | 0.205 | 0.336 | 0.199 | 0.327 | 0.199 | 0.328 | 0.198 | 0.328 | 0.198 | 0.328 | 0.339 | 0.458 | 0.207 | 0.334 |
| | 336 | 0.193 | 0.323 | 0.195 | 0.325 | 0.201 | 0.347 | 0.211 | 0.349 | 0.224 | 0.351 | 0.220 | 0.351 | 0.231 | 0.360 | 0.215 | 0.347 | 0.372 | 0.491 | 0.230 | 0.356 |
| | 720 | 0.205 | 0.330 | 0.198 | 0.330 | 0.215 | 0.358 | 0.231 | 0.373 | 0.236 | 0.365 | 0.218 | 0.353 | 0.219 | 0.353 | 0.219 | 0.353 | 0.322 | 0.467 | 0.242 | 0.370 |
| Avg. | | 0.151 | 0.250 | 0.161 | 0.258 | 0.166 | 0.266 | 0.175 | 0.275 | 0.169 | 0.269 | 0.167 | 0.267 | 0.185 | 0.287 | 0.188 | 0.296 | 0.287 | 0.418 | 0.173 | 0.272 |

Table 9. Multivariate time series forecasting results.

| Dataset | $L_y$ | ProSAR MSE | ProSAR MAE | AutoTCL MSE | AutoTCL MAE | FreRA MSE | FreRA MAE | PPT MSE | PPT MAE | TimesURL MSE | TimesURL MAE | InfoTS MSE | InfoTS MAE | TS2Vec MSE | TS2Vec MAE | TNC MSE | TNC MAE | TS–TCC MSE | TS–TCC MAE | CoST MSE | CoST MAE |
|---|---|---|---|---|---|---|---|---|---|---|---|---|---|---|---|---|---|---|---|---|---|
| ETTh₁ | 24 | 0.381 | 0.432 | 0.389 | 0.439 | 0.386 | 0.437 | 0.482 | 0.511 | 0.494 | 0.518 | 0.564 | 0.520 | 0.599 | 0.534 | 0.708 | 0.592 | 0.516 | 0.508 | 0.386 | 0.429 |
| | 48 | 0.438 | 0.469 | 0.447 | 0.477 | 0.442 | 0.471 | 0.552 | 0.561 | 0.539 | 0.557 | 0.607 | 0.553 | 0.629 | 0.555 | 0.749 | 0.619 | 0.644 | 0.579 | 0.437 | 0.464 |
| | 168 | 0.603 | 0.564 | 0.615 | 0.574 | 0.628 | 0.579 | 0.698 | 0.625 | 0.680 | 0.619 | 0.746 | 0.638 | 0.755 | 0.636 | 0.884 | 0.699 | 0.678 | 0.619 | 0.643 | 0.582 |
| | 336 | 0.780 | 0.657 | 0.802 | 0.671 | 0.815 | 0.674 | 0.916 | 0.722 | 0.891 | 0.713 | 0.904 | 0.722 | 0.907 | 0.717 | 1.020 | 0.768 | 0.967 | 0.754 | 0.812 | 0.679 |
| | 720 | 0.925 | 0.708 | 1.028 | 0.789 | 0.961 | 0.761 | 1.102 | 0.832 | 1.051 | 0.818 | 1.098 | 0.811 | 1.048 | 0.790 | 1.157 | 0.830 | 0.935 | 0.715 | 0.970 | 0.771 |
| ETTh₂ | 24 | 0.345 | 0.438 | 0.337 | 0.433 | 0.405 | 0.472 | 0.517 | 0.525 | 0.524 | 0.530 | 0.383 | 0.462 | 0.398 | 0.461 | 0.612 | 0.595 | 0.782 | 0.666 | 0.447 | 0.502 |
| | 48 | 0.591 | 0.582 | 0.572 | 0.576 | 0.695 | 0.688 | 0.681 | 0.639 | 0.679 | 0.631 | 0.567 | 0.582 | 0.578 | 0.573 | 0.840 | 0.716 | 1.357 | 0.881 | 0.699 | 0.637 |
| | 168 | 1.489 | 0.955 | 1.470 | 0.947 | 1.783 | 1.042 | 1.812 | 1.048 | 1.847 | 1.052 | 1.789 | 1.048 | 1.901 | 1.065 | 2.359 | 1.213 | 4.318 | 1.728 | 1.549 | 0.982 |
| | 336 | 1.715 | 1.028 | 1.685 | 1.027 | 1.962 | 1.099 | 2.281 | 1.213 | 2.129 | 1.182 | 2.120 | 1.161 | 2.304 | 1.215 | 2.782 | 1.349 | 2.097 | 1.145 | 1.749 | 1.042 |
| | 720 | 1.926 | 1.094 | 1.890 | 1.092 | 2.142 | 1.166 | 2.352 | 1.217 | 2.391 | 1.233 | 2.511 | 1.316 | 2.650 | 1.373 | 2.753 | 1.394 | 2.047 | 1.127 | 1.971 | 1.092 |
| ETTm₁ | 24 | 0.247 | 0.327 | 0.256 | 0.339 | 0.279 | 0.357 | 0.362 | 0.434 | 0.387 | 0.461 | 0.391 | 0.408 | 0.443 | 0.436 | 0.522 | 0.472 | 0.403 | 0.455 | 0.246 | 0.329 |
| | 48 | 0.324 | 0.384 | 0.339 | 0.396 | 0.376 | 0.420 | 0.438 | 0.547 | 0.450 | 0.558 | 0.503 | 0.475 | 0.582 | 0.515 | 0.695 | 0.567 | 0.618 | 0.552 | 0.331 | 0.386 |
| | 96 | 0.367 | 0.415 | 0.376 | 0.422 | 0.405 | 0.443 | 0.562 | 0.579 | 0.555 | 0.574 | 0.537 | 0.503 | 0.622 | 0.549 | 0.731 | 0.595 | 0.607 | 0.572 | 0.378 | 0.419 |
| | 288 | 0.452 | 0.481 | 0.464 | 0.484 | 0.501 | 0.510 | 0.674 | 0.641 | 0.657 | 0.579 | 0.653 | 0.579 | 0.709 | 0.609 | 0.818 | 0.649 | 0.722 | 0.638 | 0.472 | 0.486 |
| | 672 | 0.591 | 0.562 | 0.608 | 0.566 | 0.665 | 0.605 | 0.773 | 0.698 | 0.754 | 0.690 | 0.757 | 0.642 | 0.786 | 0.655 | 0.932 | 0.712 | 0.708 | 0.601 | 0.620 | 0.574 |
| Elec. | 24 | 0.136 | 0.240 | 0.153 | 0.250 | 0.162 | 0.258 | 0.193 | 0.285 | 0.182 | 0.276 | 0.255 | 0.350 | 0.287 | 0.374 | 0.354 | 0.423 | 0.379 | 0.561 | 0.136 | 0.242 |
| | 48 | 0.149 | 0.252 | 0.167 | 0.260 | 0.176 | 0.269 | 0.208 | 0.307 | 0.199 | 0.291 | 0.279 | 0.368 | 0.307 | 0.388 | 0.376 | 0.438 | 0.453 | 0.600 | 0.153 | 0.258 |
| | 168 | 0.165 | 0.272 | 0.179 | 0.275 | 0.183 | 0.278 | 0.213 | 0.314 | 0.205 | 0.303 | 0.302 | 0.385 | 0.332 | 0.407 | 0.402 | 0.456 | 0.575 | 0.616 | 0.175 | 0.275 |
| | 336 | 0.187 | 0.291 | 0.199 | 0.297 | 0.209 | 0.305 | 0.238 | 0.341 | 0.223 | 0.326 | 0.320 | 0.399 | 0.349 | 0.420 | 0.417 | 0.466 | 0.637 | 0.633 | 0.196 | 0.296 |
| WTH | 24 | 0.287 | 0.360 | 0.302 | 0.364 | 0.293 | 0.354 | 0.313 | 0.364 | 0.315 | 0.367 | 0.316 | 0.369 | 0.307 | 0.363 | 0.320 | 0.373 | 0.356 | 0.463 | 0.298 | 0.360 |
| | 48 | 0.348 | 0.405 | 0.361 | 0.412 | 0.355 | 0.406 | 0.372 | 0.412 | 0.377 | 0.418 | 0.381 | 0.420 | 0.374 | 0.418 | 0.380 | 0.421 | 0.429 | 0.500 | 0.359 | 0.411 |
| | 168 | 0.442 | 0.477 | 0.455 | 0.484 | 0.464 | 0.491 | 0.478 | 0.493 | 0.485 | 0.498 | 0.490 | 0.501 | 0.491 | 0.501 | 0.479 | 0.495 | 0.511 | 0.550 | 0.464 | 0.491 |
| | 336 | 0.471 | 0.501 | 0.487 | 0.505 | 0.498 | 0.517 | 0.512 | 0.518 | 0.519 | 0.524 | 0.532 | 0.527 | 0.502 | 0.507 | 0.505 | 0.514 | 0.575 | 0.584 | 0.497 | 0.517 |
| | 720 | 0.513 | 0.512 | 0.508 | 0.519 | 0.536 | 0.543 | 0.548 | 0.543 | 0.541 | 0.538 | 0.545 | 0.577 | 0.519 | 0.552 | 0.533 | 0.542 | 0.556 | 0.525 | 0.498 | 0.508 |
| Avg. | | 0.561 | 0.507 | 0.571 | 0.515 | 0.620 | 0.537 | 0.700 | 0.587 | 0.691 | 0.585 | 0.714 | 0.781 | 0.750 | 0.601 | 0.868 | 0.653 | 0.895 | 0.689 | 0.587 | 0.521 |

*Table 10.* Classification result of the UEA dataset.

| Dataset | ProSAR | AutoTCL | FreRA | PPT | TimesURL | InfoTS | TS2Vec | TNC | TS−TCC |
|---|---|---|---|---|---|---|---|---|---|
| ArticularyWordRecognition | **0.993** | 0.983 | 0.990 | 0.987 | 0.990 | **0.993** | 0.987 | 0.973 | 0.953 |
| AtrialFibrillation | **0.600** | 0.467 | 0.467 | 0.400 | 0.400 | 0.267 | 0.200 | 0.133 | 0.267 |
| BasicMotions | **1.000** | **1.000** | **1.000** | **1.000** | **1.000** | **1.000** | 0.975 | 0.975 | **1.000** |
| CharacterTrajectories | **0.995** | 0.976 | 0.991 | 0.990 | 0.990 | 0.987 | **0.995** | 0.967 | 0.985 |
| Cricket | **1.000** | **1.000** | **1.000** | **1.000** | **1.000** | **1.000** | 0.972 | 0.958 | 0.917 |
| DuckDuckGeese | 0.680 | 0.700 | **0.760** | 0.680 | 0.720 | 0.600 | 0.680 | 0.460 | 0.380 |
| EigenWorms | **0.901** | **0.901** | 0.863 | 0.847 | 0.870 | 0.748 | 0.847 | 0.840 | 0.779 |
| Epilepsy | 0.978 | 0.978 | **0.993** | 0.978 | 0.978 | **0.993** | 0.964 | 0.957 | 0.957 |
| ERing | 0.953 | 0.944 | 0.919 | 0.944 | **0.985** | 0.953 | 0.874 | 0.852 | 0.904 |
| EthanolConcentration | **0.354** | **0.354** | 0.323 | 0.346 | 0.304 | 0.323 | 0.308 | 0.297 | 0.285 |
| FaceDetection | 0.585 | 0.581 | 0.581 | **0.664** | 0.608 | 0.525 | 0.501 | 0.536 | 0.544 |
| FingerMovements | **0.660** | 0.640 | 0.610 | 0.620 | **0.660** | 0.620 | 0.480 | 0.470 | 0.460 |
| HandMovementDirection | 0.473 | 0.432 | **0.514** | 0.432 | 0.432 | **0.514** | 0.338 | 0.324 | 0.243 |
| Handwriting | 0.572 | 0.384 | **0.593** | 0.242 | 0.462 | 0.554 | 0.515 | 0.249 | 0.498 |
| Heartbeat | **0.824** | 0.785 | 0.785 | 0.766 | 0.746 | 0.771 | 0.683 | 0.746 | 0.751 |
| JapaneseVowels | 0.984 | 0.984 | 0.965 | 0.984 | **0.989** | 0.986 | 0.984 | 0.978 | 0.930 |
| Libras | 0.911 | 0.833 | 0.911 | 0.889 | **0.922** | 0.889 | 0.867 | 0.817 | 0.822 |
| LSST | **0.635** | 0.554 | 0.494 | 0.595 | 0.602 | 0.593 | 0.537 | 0.595 | 0.474 |
| MotorImagery | 0.620 | 0.570 | 0.550 | 0.610 | **0.680** | 0.610 | 0.510 | 0.500 | 0.610 |
| NATOPS | 0.939 | 0.944 | 0.900 | 0.917 | **0.961** | 0.939 | 0.928 | 0.911 | 0.822 |
| PEMS-SF | 0.821 | **0.838** | 0.746 | 0.821 | 0.821 | 0.757 | 0.682 | 0.699 | 0.734 |
| PenDigits | **0.989** | 0.984 | 0.973 | 0.984 | **0.989** | **0.989** | **0.989** | 0.979 | 0.974 |
| PhonemeSpectra | 0.237 | 0.218 | **0.274** | 0.233 | 0.237 | 0.233 | 0.233 | 0.207 | 0.252 |
| RacketSports | 0.862 | **0.914** | 0.888 | 0.862 | 0.862 | 0.829 | 0.855 | 0.776 | 0.816 |
| SelfRegulationSCP1 | 0.891 | 0.891 | 0.908 | **0.932** | 0.908 | 0.887 | 0.812 | 0.799 | 0.823 |
| SelfRegulationSCP2 | **0.622** | 0.578 | **0.622** | 0.561 | 0.600 | 0.527 | 0.578 | 0.550 | 0.533 |
| SpokenArabicDigits | **0.988** | 0.925 | 0.984 | 0.981 | 0.985 | **0.988** | 0.932 | 0.934 | 0.970 |
| StandWalkJump | 0.467 | 0.533 | **0.667** | 0.467 | 0.467 | 0.467 | 0.467 | 0.400 | 0.333 |
| UWaveGestureLibrary | 0.903 | 0.893 | 0.900 | 0.856 | **0.919** | 0.906 | 0.884 | 0.753 | 0.753 |
| InsectWingbeat | **0.488** | **0.488** | 0.462 | 0.472 | 0.473 | 0.472 | 0.466 | 0.469 | 0.264 |
| Avg. ACC | **0.764** | 0.742 | 0.754 | 0.735 | 0.752 | 0.730 | 0.704 | 0.670 | 0.668 |
| Avg. RANK | **1.867** | 3.067 | 2.900 | 3.200 | 2.233 | 3.133 | 4.367 | 5.500 | 5.367 |

**Implementation and Evaluation Protocol.** For MHCCL, we utilized the official open-source implementation. Regarding AimTS, since the official code is not publicly available, we reproduced the method based on the algorithmic details provided in the original paper. Crucially, to ensure a fair assessment of the intrinsic representation quality across all baselines, we strictly adhered to the standard evaluation protocol established by TS2Vec (Yue et al., 2022) and adopted by AutoTCL. This protocol involves training a task-specific predictor (e.g., SVM or Ridge Regression) on top of *frozen* representations from the pre-trained encoder. We note that while the original AimTS study utilizes a full fine-tuning paradigm, evaluating it under this standardized frozen protocol isolates the quality of the learned representations from the benefits of supervised fine-tuning.

**Analysis of Results.** The comparative results demonstrate that ProSAR consistently outperforms both prototype-based baselines across forecasting and classification tasks:

- **Forecasting Tasks:** In univariate forecasting, ProSAR achieves an average MSE of **0.151**, surpassing MHCCL (0.176) by approximately 14% and AimTS (0.171) by over 11%. This superiority is maintained in the multivariate setting, where ProSAR achieves an MSE of **0.561**, compared to 0.627 for MHCCL and 0.641 for AimTS. These results indicate that ProSAR's strategy of using prototypes to actively guide upstream augmentation preserves critical temporal dynamics more effectively than methods that use prototypes solely as downstream loss constraints.

- **Classification Tasks:** On the UEA classification benchmarks, ProSAR achieves a state-of-the-art average accuracy of **0.764**, markedly outperforming MHCCL (0.705) and AimTS (0.715). The performance gap with AimTS under the frozen protocol highlights that while AimTS relies heavily on task-specific fine-tuning to adjust its features, ProSAR is capable of learning highly discriminative representations intrinsically during the pre-training stage.

## C.4. Ablation Studies: Detailed Results

The main paper summarizes the average forecasting performance of ablation studies. Detailed per-dataset results for these specific ablations (**ProSAR (Full)**, **w/o DTW-Seg**, **w/o STFT**, **w/o Dual-Proto**, **w/o Clustering**, and **w/o Decoding**) are provided in Table 11. These results underscore the critical contribution of each component within the ProSAR framework, particularly highlighting the efficacy of the prototype-guided segmentation, frequency-domain alignment, and the closed-loop dynamic prototype refinement strategy.

*Table 11.* Detailed ablation study results for univariate forecasting tasks across different prediction lengths ($L_y$). **Note:** The per-dataset mean is computed by averaging the reported horizons within each dataset block, and the final Avg. is the mean of the per-dataset means (ETTh$_1$, ETTh$_2$, ETTm$_1$, Elec., WTH).

| Dataset | $L_y$ | ProSAR (Full) | | w/o DTW-Seg | | w/o STFT | | w/o Dual-Proto | | w/o Clustering | | w/o Decoding | |
|---|---|---|---|---|---|---|---|---|---|---|---|---|---|
| | | MSE | MAE | MSE | MAE | MSE | MAE | MSE | MAE | MSE | MAE | MSE | MAE |
| ETTh$_1$ | 24 | **0.035** | **0.142** | 0.044 | 0.157 | 0.045 | 0.160 | 0.037 | 0.147 | 0.046 | 0.160 | 0.039 | 0.150 |
| | 48 | **0.051** | **0.169** | 0.064 | 0.187 | 0.066 | 0.190 | 0.054 | 0.175 | 0.067 | 0.190 | 0.056 | 0.178 |
| | 168 | **0.074** | **0.205** | 0.093 | 0.227 | 0.096 | 0.231 | 0.078 | 0.212 | 0.097 | 0.231 | 0.081 | 0.216 |
| | 336 | **0.082** | **0.221** | 0.104 | 0.244 | 0.106 | 0.249 | 0.087 | 0.229 | 0.107 | 0.249 | 0.090 | 0.233 |
| | 720 | **0.100** | **0.253** | 0.126 | 0.280 | 0.129 | 0.285 | 0.106 | 0.262 | 0.130 | 0.285 | 0.110 | 0.267 |
| ETTh$_2$ | 24 | **0.075** | **0.203** | 0.093 | 0.223 | 0.096 | 0.228 | 0.080 | 0.208 | 0.098 | 0.217 | 0.083 | 0.209 |
| | 48 | **0.111** | **0.250** | 0.138 | 0.274 | 0.142 | 0.280 | 0.118 | 0.256 | 0.144 | 0.267 | 0.123 | 0.258 |
| | 168 | **0.166** | **0.313** | 0.206 | 0.343 | 0.212 | 0.351 | 0.176 | 0.321 | 0.216 | 0.335 | 0.184 | 0.323 |
| | 336 | **0.178** | **0.338** | 0.221 | 0.371 | 0.228 | 0.379 | 0.189 | 0.346 | 0.231 | 0.361 | 0.198 | 0.349 |
| | 720 | **0.182** | **0.343** | 0.222 | 0.374 | 0.232 | 0.382 | 0.192 | 0.349 | 0.236 | 0.365 | 0.202 | 0.351 |
| ETTm$_1$ | 24 | **0.014** | **0.083** | 0.017 | 0.089 | 0.017 | 0.091 | 0.015 | 0.085 | 0.017 | 0.090 | 0.015 | 0.086 |
| | 48 | **0.024** | **0.114** | 0.028 | 0.122 | 0.029 | 0.125 | 0.026 | 0.117 | 0.030 | 0.123 | 0.026 | 0.119 |
| | 96 | **0.035** | **0.141** | 0.041 | 0.151 | 0.043 | 0.155 | 0.037 | 0.145 | 0.043 | 0.152 | 0.038 | 0.147 |
| | 288 | **0.066** | **0.195** | 0.078 | 0.209 | 0.081 | 0.215 | 0.071 | 0.200 | 0.082 | 0.211 | 0.072 | 0.203 |
| | 672 | **0.088** | **0.223** | 0.101 | 0.239 | 0.105 | 0.245 | 0.091 | 0.229 | 0.108 | 0.241 | 0.094 | 0.232 |
| Elec. | 24 | **0.237** | **0.257** | 0.245 | 0.276 | 0.256 | 0.281 | 0.242 | 0.266 | 0.249 | 0.274 | 0.245 | 0.268 |
| | 48 | **0.278** | **0.284** | 0.288 | 0.305 | 0.300 | 0.311 | 0.284 | 0.294 | 0.292 | 0.303 | 0.288 | 0.296 |
| | 168 | **0.383** | **0.352** | 0.396 | 0.378 | 0.414 | 0.385 | 0.391 | 0.364 | 0.402 | 0.376 | 0.396 | 0.367 |
| | 336 | **0.452** | **0.420** | 0.471 | 0.449 | 0.494 | 0.459 | 0.463 | 0.432 | 0.477 | 0.447 | 0.471 | 0.437 |
| WTH | 24 | **0.091** | **0.208** | 0.110 | 0.227 | 0.114 | 0.231 | 0.099 | 0.213 | 0.108 | 0.223 | 0.106 | 0.218 |
| | 48 | **0.131** | **0.256** | 0.159 | 0.279 | 0.164 | 0.285 | 0.143 | 0.262 | 0.156 | 0.274 | 0.152 | 0.269 |
| | 168 | **0.180** | **0.309** | 0.218 | 0.337 | 0.225 | 0.344 | 0.196 | 0.317 | 0.214 | 0.331 | 0.209 | 0.324 |
| | 336 | **0.193** | **0.323** | 0.234 | 0.352 | 0.241 | 0.359 | 0.210 | 0.331 | 0.230 | 0.346 | 0.224 | 0.339 |
| | 720 | **0.205** | **0.330** | 0.249 | 0.360 | 0.256 | 0.366 | 0.222 | 0.337 | 0.242 | 0.351 | 0.239 | 0.345 |
| Avg. | | **0.151** | **0.250** | 0.172 | 0.272 | 0.178 | 0.278 | 0.158 | 0.257 | 0.175 | 0.270 | 0.164 | 0.261 |

## C.5. Network Architecture Choices

ProSAR employs distinct encoder architectures for its forecasting and classification tasks, primarily to ensure fair comparisons with state-of-the-art methods by using established backbones relevant to each task domain.

For time series forecasting, the encoder architecture in ProSAR is designed to mirror that of CoST (Woo et al., 2022). This involves a multi-layer dilated Convolutional Neural Network (CNN) as its backbone and we remove the seasonal feature disentangler module. For time series classification experiments, ProSAR adopts the encoder architecture from TS2Vec (Yue et al., 2022). This allows for a consistent comparison framework with other methods evaluated using this common backbone for classification benchmarks. The main forecasting results presented in Section 4.1 of the paper utilize the ProSAR framework primarily with the CoST-derived backbone. Table 12 serves as an illustrative results for univariate forecasting tasks, comparing the performance of ProSAR's core methodology when integrated with the CoST backbone

versus the TS2Vec backbone. As indicated by the metrics, a slight decrease in performance is observed when the TS2Vec backbone is used for these forecasting tasks instead of the CoST backbone, highlighting the effective synergy of the selected CoST-based architecture for ProSAR's forecasting setup. This also demonstrates the adaptability of the ProSAR framework's core components, which are potentially compatible with other encoders, with such integrations being subjects for future exploration.

*Table 12.* Univariate forecasting results with different backbones.

| Method | ETTh$_1$ | ETTh$_2$ | ETTm$_1$ | Elec. | WTH |
|---|---|---|---|---|---|
| ProSAR (CoST backbone) | 0.068 / 0.198 | 0.142 / 0.289 | 0.045 / 0.151 | 0.338 / 0.328 | 0.160 / 0.285 |
| ProSAR (TS2Vec backbone) | 0.072 / 0.208 | 0.150 / 0.305 | 0.048 / 0.160 | 0.370 / 0.350 | 0.167 / 0.301 |

## C.6. Parameter Sensitivity Analysis

This section explores ProSAR's sensitivity to key hyperparameters. We vary parameters of loss component weights ($\lambda_{\text{intra}}, \lambda_{\text{inter}}$) over reasonable ranges and observe their impact on performance on the ETTh1 dataset. Figure 5 presents this analysis, showing Mean Squared Error (MSE) and Mean Absolute Error (MAE) versus hyperparameter values. This analysis helps understand the robustness of ProSAR and provides guidance for hyperparameter selection.

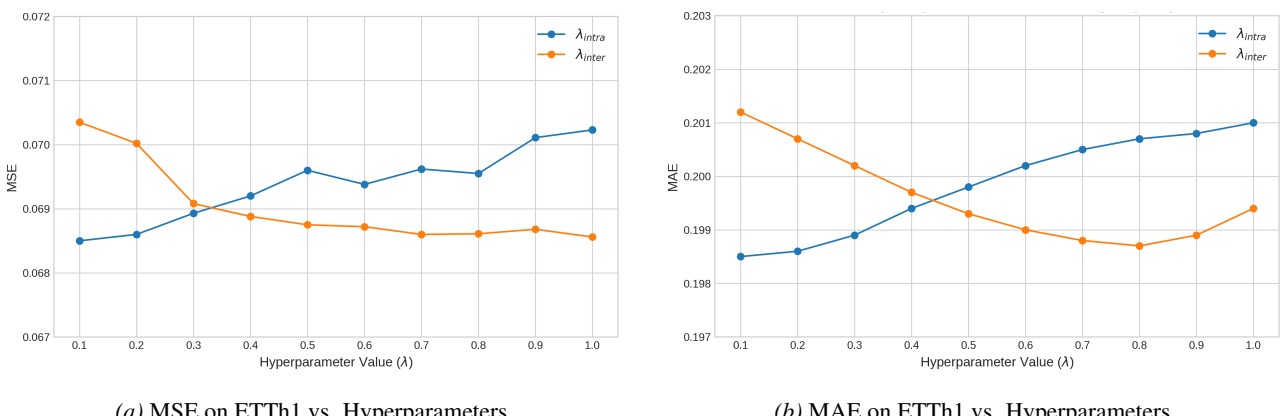

*(a)* MSE on ETTh1 vs. Hyperparameters          *(b)* MAE on ETTh1 vs. Hyperparameters

*Figure 5.* Sensitivity analysis of ProSAR on the ETTh1 dataset, illustrating the impact of key hyperparameter variations on (a) Mean Squared Error (MSE) and (b) Mean Absolute Error (MAE).

## C.7. Details regarding Figure 2: Timestamp-wise Mutual Information

Directly estimating the mutual information between an entire multivariate time series and its class label, i.e., $MI(x; y)$ or $MI(v; y)$ where $v = T(x)$, is non-trivial due to the curse of dimensionality. To make the estimation tractable and to visualize how semantic information is distributed along time, in Figure 2 we report a *timestamp-wise* mutual information curve. Concretely, for each timestamp $t \in \{1, \ldots, L\}$, we compute $MI(v_t; y)$, where $v_t \in \mathbb{R}^D$ denotes the $D$-dimensional observation of the augmented view at time $t$ (and similarly $MI(x_t; y)$ for the raw input). This yields a length-$L$ curve that visualizes how much label-relevant information is preserved across timestamps under different augmentation functions. In our Figure 2, the dataset is the UEA `LSST` time-series classification benchmark.

**Estimator implementation.** At each timestamp $t$, we estimate $MI(v_t; y)$ using the kNN-based mutual information estimator implemented in `sklearn.feature_selection.mutual_info_classif`. Since $v_t$ is $D$-dimensional, `mutual_info_classif` returns a per-feature estimate $\widehat{MI}(v_{t,d}; y)$ for each dimension $d \in \{1, \ldots, D\}$; we then report their mean, i.e., $\widehat{MI}(v_t; y) \triangleq \frac{1}{D} \sum_{d=1}^{D} \widehat{MI}(v_{t,d}; y)$, and analogously for $\widehat{MI}(x_t; y)$. We set the neighborhood size to `n_neighbors = k` (default $k = 3$ unless specified), clipped to be $1 \le k \le N - 1$ where $N$ is the number of instances.

Estimating mutual information at each timestamp only involves a low-dimensional random vector in $\mathbb{R}^D$, which substantially alleviates the dimensionality issue compared to estimating $MI(x; y)$ on the full sequence. We emphasize that we do not intend to suggest that any single timestamp alone is fully representative of the underlying semantics of the entire sequence. Instead, the plot serves as a diagnostic tool to show how the informative content varies over time, and how different

view-generation strategies preserve label-relevant information at different temporal locations. In particular, a method whose $MI(v_t; y)$ curve stays close to $MI(x_t; y)$ across $t$ indicates stronger semantic retention in the generated views, which is consistent with the motivation of ProSAR to progressively approach $MI(x; y)$ through the prototype-refinement feedback loop.

## C.8. Convergence Analysis

To demonstrate the training stability of ProSAR, we plot the training loss curves for five representative datasets. Figure 6 shows the evolution of the total loss $\mathcal{L}_{\text{total}}$, as well as its main components $L_{\text{intra}}$ and $L_{\text{inter}}$ ($L_{\text{inter\_inst}}$ and $L_{\text{inter\_proto}}$), over training epochs. The curves across these diverse datasets indicate that ProSAR generally converges smoothly.

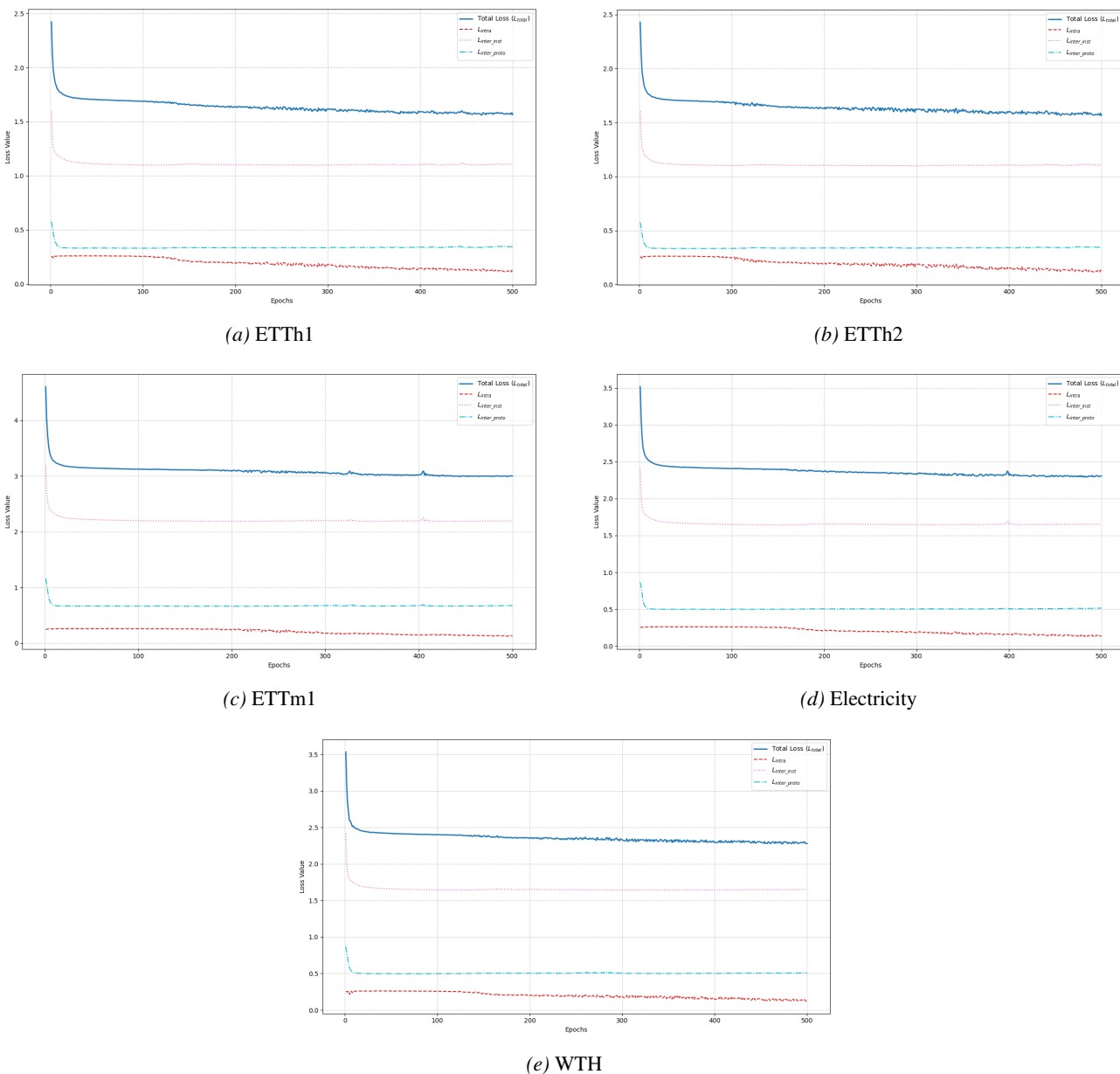

*Figure 6.* Convergence of ProSAR training losses on five representative datasets: ETTh1, ETTh2, ETTm1, Electricity, and WTH. Each subfigure shows the total loss ($\mathcal{L}_{\text{total}}$) and its primary components over epochs.

## C.9. Computational Efficiency and Scalability Analysis

We provide a more detailed analysis of the computational efficiency and scalability of ProSAR. The additional overhead of ProSAR mainly comes from prototype-guided alignment, latent clustering, mutual-information-based refinement, and frequency-domain augmentation. These operations are only used during self-supervised pretraining and introduce no extra computational cost during downstream inference.

**Scalability with respect to training data size.** We first evaluate the training time when varying the ratio of training data while keeping the other settings fixed. As shown in Table 13, the average epoch time increases from 228 ms at 25% data to 848 ms at 100% data, showing an approximately linear growth trend with respect to the amount of training data.

*Table 13.* Scalability with respect to training data size. We report the average training time per epoch under different training-set ratios.

| Train ratio | Epoch time (ms) | Relative time |
|---|---|---|
| 25% | 228 | 0.27× |
| 50% | 421 | 0.50× |
| 75% | 672 | 0.79× |
| 100% | 848 | 1.00× |

**Scalability with respect to sequence length.** We further evaluate the training time under different input lengths. As shown in Table 14, when the input length increases from 64 to 512, the average epoch time increases from 688 ms to 934 ms. The growth is moderate rather than dramatic, suggesting that the prototype-guided alignment and refinement modules do not become the dominant bottleneck as the sequence length increases.

*Table 14.* Scalability with respect to sequence length. We report the average training time per epoch under different input lengths.

| Input length | Epoch time (ms) | Relative time |
|---|---|---|
| 64 | 688 | 1.00× |
| 128 | 780 | 1.13× |
| 256 | 848 | 1.23× |
| 512 | 934 | 1.43× |

**Comparison with representative baselines.** We also compare the practical training time of ProSAR with representative time-series contrastive learning baselines. As reported in Table 15, ProSAR has an average epoch time of 0.848 s, which is higher than TimesURL but lower than AutoTCL and FreRA. This indicates that although ProSAR introduces additional prototype-guided operations, its practical runtime remains moderate among recent self-supervised time-series methods.

*Table 15.* Average training time per epoch compared with representative baselines.

| Method | Epoch time (s) |
|---|---|
| TimesURL | 0.583 |
| ProSAR | 0.848 |
| FreRA | 1.296 |
| AutoTCL | 1.552 |

**Module-level runtime breakdown.** To identify the source of the runtime overhead, we further provide a module-level breakdown in Table 16. The dominant cost comes from the main optimization step, where the loss computation and backward propagation take 690.972 ms and account for 81.48% of the epoch time. In contrast, the DTW-related overhead remains small: DTW alignment and matching together take 24.304 ms, accounting for only 2.87%. FINCH clustering takes 72.452 ms, accounting for 8.54%. Other modules, including KMeans with prototype fusion, MI refinement, and frequency augmentation, also account for relatively small portions of the total runtime. Therefore, the additional prototype-guided operations are not the dominant computational bottleneck of ProSAR.

*Table 16.* Module-level runtime breakdown of ProSAR. The DTW overhead is reported as the sum of alignment and matching. Lower is better.

| Component | Avg. ms / epoch | Percentage |
|---|---|---|
| Total epoch | 848.000 | 100.00% |
| Loss + backward | 690.972 | 81.48% |
| DTW align + match | 24.304 | 2.87% |
| FINCH clustering | 72.452 | 8.54% |
| KMeans + proto. fusion | 28.708 | 3.39% |
| MI refinement | 19.612 | 2.31% |
| Frequency augmentation | 12.388 | 1.46% |

# D. Visualizations and Qualitative Analysis

This section provides visual examples and qualitative analysis of ProSAR's key mechanisms. These visualizations aim to offer a clearer understanding of how ProSAR identifies semantic content and generates diverse, yet semantically consistent, augmented views.

## D.1. Visualizations of Patterns and Alignment

In this section, we provide further visual evidence to verify that the learned prototypes capture meaningful temporal dynamics and that the alignment mechanism functions as intended.

**Visual Evidence of Canonical Waveform Patterns.** Qualitative analysis reveals that learned time-domain prototypes evolve into distinct, canonical waveform structures rather than random noise. As illustrated in Figure 7, the learned prototypes (e.g., Proto 1, Proto 3, Proto 12) exhibit clear, repetitive cyclical patterns. Frequency analysis confirms these prototypes capture the dominant periodicity of the dataset (approximately 8 time steps for ETTm1), validating that they successfully distill high-level temporal semantics and serve as representative templates for their respective clusters.

**Semantic Verification via Explicit Alignment Paths.** The semantic validity of these prototypes is further verified by the DTW alignment mechanism, which physically distinguishes structural content from noise. We provide visualizations of the DTW alignment matrices and cumulative cost paths in Figure 8. These plots explicitly show the model aligning input segments to their assigned prototypes (e.g., Seg 3 aligned to Proto 1) along diagonal paths, which indicate strong structural correspondence. Crucially, the warped prototypes (orange dashed lines) closely track the dominant trends and cycles of the input segments (blue lines), demonstrating that the model uses the prototype as a physical ground truth to identify the semantic core ($x_S$) while treating deviations as non-semantic variations.

## D.2. Visualization of Latent Space Prototypes

To understand the structure of the learned semantic anchors in the latent space, Figure 9 presents two t-SNE visualizations of the latent prototypes $\{p_k^z\}$ derived from the Cricket dataset after training ProSAR. An effective set of latent prototypes should ideally be well-distributed in the latent space, indicating that they capture diverse semantic clusters. These visualizations (from different training stages) help assess the separation and grouping of these latent concepts, which underpins the model's ability to differentiate between semantically distinct time series patterns.

## D.3. Visualizations of Semantic Segment Identification and Augmentation

Figure 10 provides a detailed, step-by-step illustration of ProSAR's prototype-guided augmentation mechanism applied to a sample time series from the Cricket dataset. This process aims to preserve core semantic content while introducing diversity, guided by the information-theoretic principles outlined in the main paper. The stages are: (a) The original input series $x$. (b) The specific time-domain prototype $p_k^x$ selected by DTW as most similar to $x$. (c) The identified semantic segment $x_S = x \odot M_x$, extracted from $x$ based on high-quality alignment with $p_k^x$. $M_x$ is the binary mask derived from DTW. (d) The original non-semantic part $x_N = x \odot (1 - M_x)$, representing regions of $x$ with poor alignment to $p_k^x$. (e) The transformed semantic segment $x_S'$. This involves DTW-guided temporal alignment normalization and controlled noise injection in the frequency domain, designed to preserve essential semantics while discarding P-irrelevant nuisance variability. (f) The perturbed non-semantic part $x_N'$. This part is heavily modified through noise and random sub-segment masking to corrupt P-irrelevant information. (g) The final augmented view $\tilde{x}$, constructed by combining the transformed semantic part

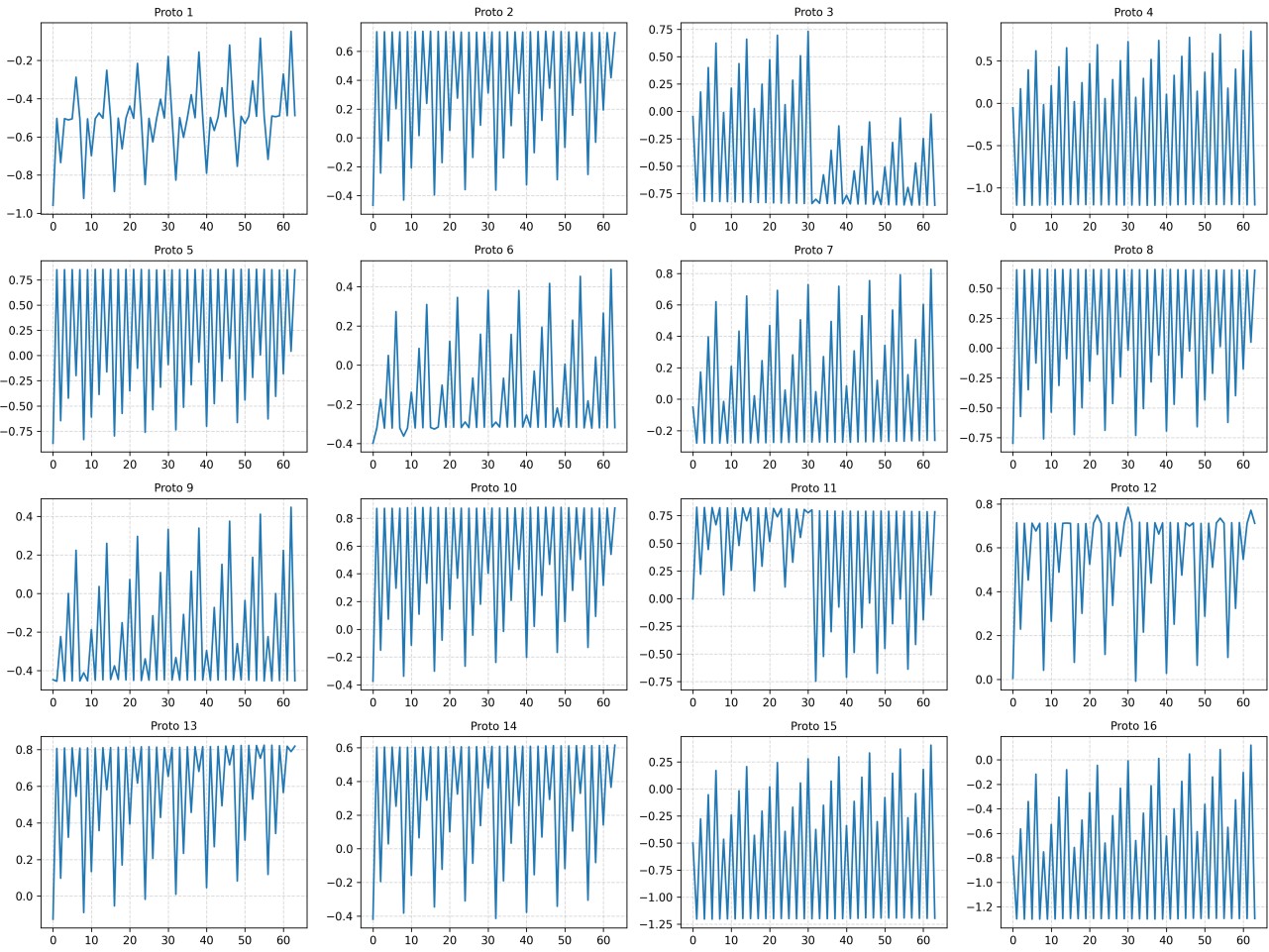

*Figure 7.* Visualization of 16 learned time-domain prototypes. The waveforms exhibit clear, distinct cyclical patterns rather than noise, demonstrating the model's ability to distill canonical temporal structures from the dataset.

and the perturbed non-semantic part: $\tilde{x} = (x'_S \odot M_x) + (x'_N \odot (1 - M_x))$. This visual walkthrough clarifies how learnable prototypes and DTW-guided segmentation enable ProSAR to generate semantically coherent and diverse augmentations.

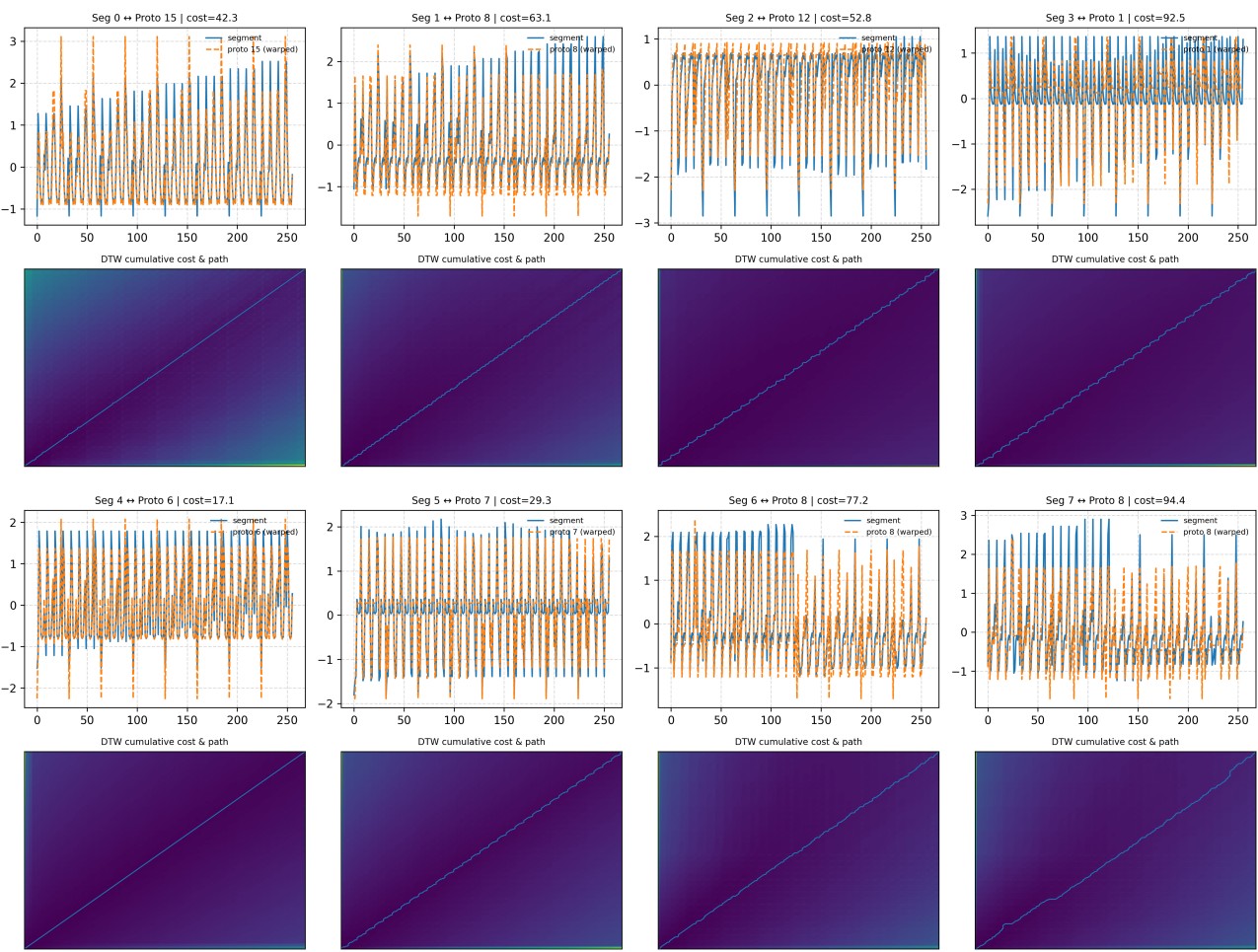

*Figure 8.* Detailed visualization of the DTW alignment mechanism. **Top Row:** Input segments (blue) overlaid with their assigned, warped prototypes (orange dashed). **Bottom Row:** Corresponding DTW cumulative cost matrices. The distinct diagonal paths in the matrices (indicating optimal alignment) and the tight visual fit between segments and warped prototypes confirm that the model is successfully identifying semantic structural correspondence.

# E. Theoretical Derivations and Proofs

This appendix provides complete and self–contained proofs for the theoretical claims in the main paper. For ease of cross–referencing we follow the numbering of the propositions and theorems used in the main text while introducing additional appendix-only labels where necessary.

**Notation recap.** $X$: raw time series; $\tilde{X} = T(X)$: augmented view; $Z = f_\theta(X)$: latent representation; $P = g(X)$: hard assignment to prototypes; $\beta \in (0, 1)$: IB trade-off parameter.

## E.1. Proof of Proposition 3.1 (Semantic Information Decomposition)

We must show $I(X; \tilde{X}) = I(P; \tilde{X}) + I(X; \tilde{X} \mid P)$.

**Step 1 (chain rule).** For any triplet $(V_1, V_2, V_3)$ the chain rule for mutual information states $I(V_1, V_2; V_3) = I(V_1; V_3) + I(V_2; V_3 \mid V_1)$. Setting $(V_1, V_2, V_3) = (P, X, \tilde{X})$ yields

$$I(P, X; \tilde{X}) = I(P; \tilde{X}) + I(X; \tilde{X} \mid P). \tag{11}$$

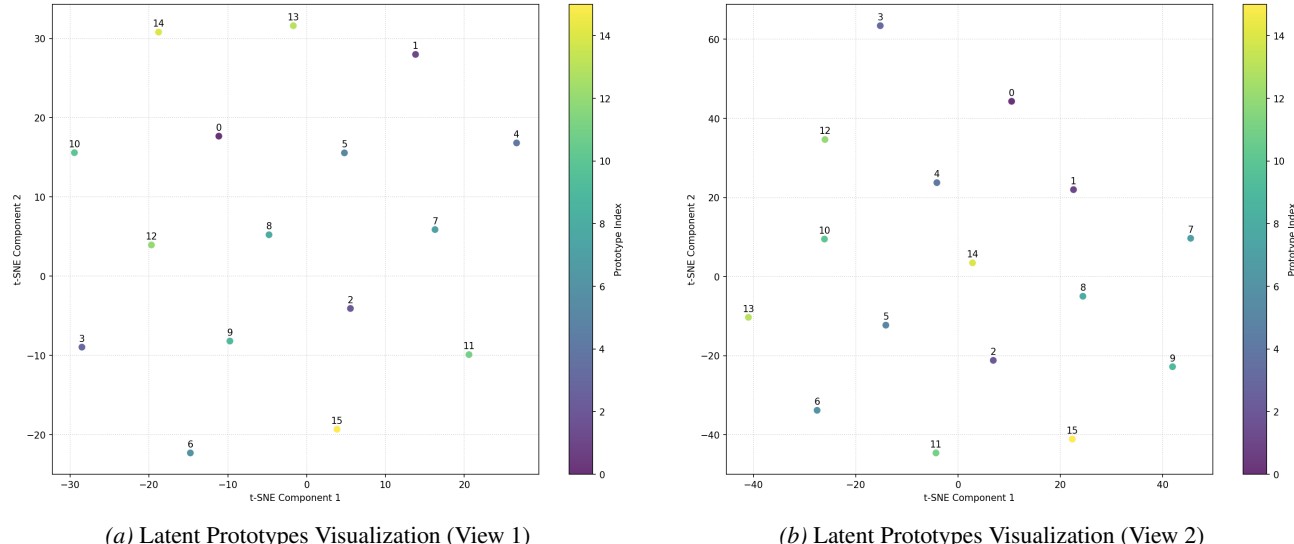

*(a)* Latent Prototypes Visualization (View 1)        *(b)* Latent Prototypes Visualization (View 2)

*Figure 9.* Visualizations of learned latent space prototypes.

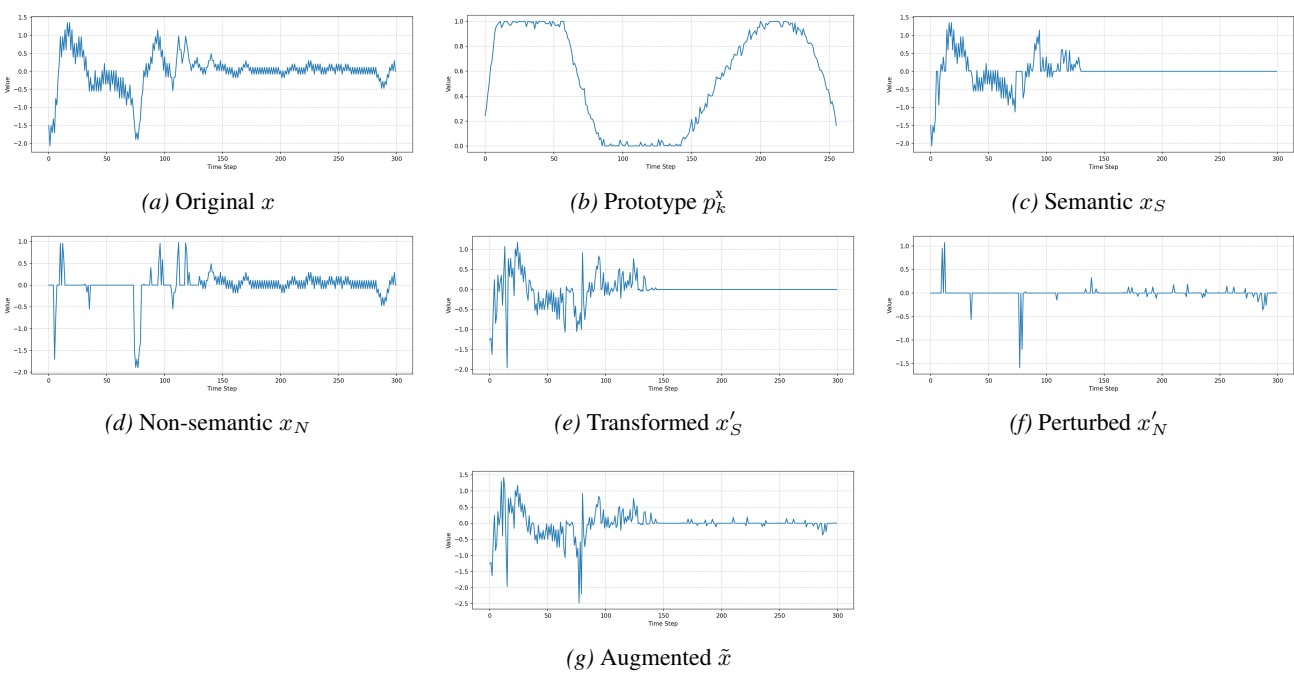

*(a)* Original $x$        *(b)* Prototype $p_k^{\mathrm{x}}$        *(c)* Semantic $x_S$

*(d)* Non-semantic $x_N$        *(e)* Transformed $x_S'$        *(f)* Perturbed $x_N'$

*(g)* Augmented $\tilde{x}$

*Figure 10.* Step-by-step visualization of ProSAR's prototype-guided semantic augmentation process (a-g).

**Step 2 (deterministic prototype map).**   Because $P = g(X)$ is deterministic, $H(P, X) = H(X) + H(P \mid X) = H(X)$, and similarly, $H(P, X \mid \tilde{X}) = H(X \mid \tilde{X}) + H(P \mid X, \tilde{X}) = H(X \mid \tilde{X})$. Hence $I(P, X; \tilde{X}) = H(X) - H(X \mid \tilde{X}) = I(X; \tilde{X})$.

**Step 3 (substitution).**   Substituting the above equality into (11) proves the desired identity.   ∎

### E.2. Proof of Proposition 3.2 (Prototype-Optimal Augmentation)

Recall the optimization objective

$$\mathcal{L}(T) = (1 - \beta)\, I(P; \tilde{X}) \ - \ \beta\, I(X; \tilde{X} \mid P), \quad 0 < \beta < 1. \tag{12}$$

**Term 1.** Because of the data–processing inequality (DPI) for the Markov chain $P \leftarrow X \rightarrow \tilde{X}$, $I(P; \tilde{X}) \leq I(P; X)$. The upper bound is attained iff $\tilde{X}$ retains every bit of information that $X$ contains about $P$, i.e. $I(P; \tilde{X}) = I(P; X)$.

**Term 2.** The conditional mutual information $I(X; \tilde{X} \mid P) \geq 0$ with equality *iff* $X$ and $\tilde{X}$ are conditionally independent given $P$ $(X \leftrightarrow P \leftrightarrow \tilde{X})$.

**Optimality conditions.** Maximizing (12) therefore requires simultaneously

(i) $I(X; \tilde{X} \mid P) \rightarrow 0$,

(ii) $I(P; \tilde{X}) \rightarrow I(P; X)$.

These two conditions are exactly those stated in the main text. ∎

### E.3. Information-Theoretic Interpretation of Prototype Updates

Prototype refinement can be viewed through the *Information Bottleneck* (IB) lens. Assume the downstream target is an (unknown) random variable $Y_{\text{task}}$ that depends on the representation $Z = f_\theta(X)$. A *good* assignment variable $P$ should

(a) retain as much information as possible about $Y_{\text{task}}$, i.e. maximise $I(P; Y_{\text{task}})$;

(b) yet be a *compressed* summary of $Z$, controlled by $I(P; Z)$.

**IB formulation.** This trade-off is formalized by

$$\max_{q(P|Z)} \ I(P; Y_{\text{task}}) \ - \ \gamma \, I(P; Z), \qquad \gamma > 0. \tag{13}$$

**Self-supervised surrogate.** Because $Y_{\text{task}}$ is unavailable in self-supervised pre-training, we adopt a common surrogate: *maximise $I(P; Z)$ while penalising the complexity of $P$*. Two equivalent forms appear in the literature:

(i) a *rate–distortion* variant $\max_q \ I(P; Z) - \lambda H(P)$ with fixed $\lambda$;

(ii) a *capacity-constrained* variant $\max_q \ I(P; Z)$ s.t. $H(P) \leq C$.

*Remark.* In PROSAR we simply fix the prototype count to $K$; this enforces a constant upper bound $H(P) \leq \log K$, so the capacity-constrained form is satisfied implicitly and we do not write $H(P)$ explicitly in later proofs.

*Remark.* In the absence of labels, keeping the original $-\gamma I(P; Z)$ term would drive $P$ to discard all information. Hence we replace the supervised fidelity term $I(P; Y_{\text{task}})$ by $I(P; Z)$ *and* moves the compression pressure to a separate regulariser such as $H(P)$ or a fixed cluster count $K$.

**On the use of prototypes as semantic proxies.** In Eq. (3) the unknown semantic variable $C$ is replaced by the prototype index $P = g(X)$. This mirrors the replacement of $Y_{\text{task}}$ by $Z$ in the information–bottleneck objective: both substitutions turn unobservable quantities into deterministic proxies that are refined during training. Because $P$ is a deterministic function of $X$, substituting $C \rightarrow P$ does not affect the mutual–information identities, while the iterative refinement loop guarantees that the quality of this proxy can only improve. Extensive experiments confirm that the learned prototypes indeed capture task–relevant semantics.

**Hard clustering as a variational IB solver.** Let the encoder outputs be i.i.d. samples from an isotropic Gaussian mixture, $p(z \mid P = k) = \mathcal{N}(\mu_k, \sigma^2 I)$ with uniform priors. The log-likelihood of data is $\log p(Z \mid \{\mu_k\}) = -\frac{1}{2\sigma^2} \sum_k \sum_{i \in C_k} \|z_i - \mu_k\|_2^2 + \text{const}$. Maximising this likelihood under *hard assignments* $q(P = k \mid z) = \mathbf{1}[k = \arg\min_j \|z - \mu_j\|]$ is precisely Lloyd's distortion minimization (15). Moreover,

$$I(P; Z) = H(Z) - H(Z \mid P) \qquad \qquad \text{(definition)}$$

$$\geq H(Z) - \frac{d}{2N} \sum_{k=1}^{K} \sum_{i \in C_k} \log\left( \tfrac{2\pi e}{d} \|z_i - \mu_k\|_2^2 \right) \quad \text{(Gaussian max–entropy bound)} \tag{14}$$

so *decreasing distortion increases a lower bound on $I(P; Z)$.* Hence any algorithm that *monotonically decreases* (15) performs a *greedy ascent* on the IB surrogate (13). This directly connects to Theorem E.1 proved next.

Having established the IB perspective, we now formalise the guarantee for *arbitrary* distortion–decreasing hard-clustering (Theorem E.1), and then instantiate it for FINCH and input-space $k$-means.

### E.4. Prototype–IB Optimality for Distortion–Decreasing Hard Clustering

Below we prove a generic result (Theorem E.1) showing that *any* hard-clustering algorithm that monotonically decreases the within–cluster $\ell_2$ distortion also monotonically increases a lower bound on $I(P; Z)$. Afterwards we instantiate the theorem for FINCH and for vanilla $k$-means applied in the input space.

**Theorem E.1** (Prototype-IB Optimality). *Let $Z = \{z_i\}_{i=1}^N \subset \mathbb{R}^d$. A hard-clustering iteration produces assignments $P^{(t)} \in [K]^N$ and centroids $\mu_k^{(t)}$. Define the within–cluster $\ell_2$ distortion*

$$D_t = \sum_{k=1}^K \sum_{i \in C_k^{(t)}} \| z_i - \mu_k^{(t)} \|_2^2, \tag{15}$$

*where $C_k^{(t)} = \{ i : P_i^{(t)} = k \}$ and $\mu_k^{(t)} = \frac{1}{|C_k^{(t)}|} \sum_{i \in C_k^{(t)}} z_i$. If the algorithm guarantees $D_{t+1} \le D_t$ for every t, then*

$$H\big(Z \,|\, P^{(t+1)}\big) \;\le\; H\big(Z \,|\, P^{(t)}\big), \qquad I\big(P^{(t+1)}; Z\big) \;\ge\; I\big(P^{(t)}; Z\big).$$

*Consequently the sequence $\{I(P^{(t)}; Z)\}_{t \ge 0}$ is monotone non-decreasing and converges at a (possibly local) minimum of $D_t$.*

*Proof.* **Upper bound on $H(Z \mid P)$.** For each cluster $k$ the empirical conditional distribution $p_k$ has covariance matrix $\Sigma_k = \frac{1}{|C_k|} \sum_{i \in C_k} (z_i - \mu_k)(z_i - \mu_k)^\top$. Maximum-entropy principle gives $h(p_k) \le \frac{1}{2} \log\big[(2\pi e)^d \det \Sigma_k\big]$. Using A.M.–G.M. inequality $\det \Sigma_k \le (\frac{1}{d} \mathrm{tr}\, \Sigma_k)^d$ and $\mathrm{tr}\, \Sigma_k = \frac{1}{|C_k|} \sum_{i \in C_k} \| z_i - \mu_k \|_2^2$, we obtain the bound

$$H(Z \mid P) \;\le\; \frac{d}{2N} \sum_k \sum_{i \in C_k} \log\Big( \tfrac{2\pi e}{d} \| z_i - \mu_k \|_2^2 \Big). \tag{16}$$

The right-hand side is a strictly increasing function of every squared distance and hence of the aggregate distortion $D_t$.

**Monotone decrease of $H(Z \mid P)$.** Because $D_{t+1} \le D_t$ by assumption, (16) implies $H(Z \mid P^{(t+1)}) \le H(Z \mid P^{(t)})$.

**Monotone ascent of $I(P; Z)$.** $H(Z)$ is dataset-constant, therefore $I(P^{(t+1)}; Z) - I(P^{(t)}; Z) = H(Z \mid P^{(t)}) - H(Z \mid P^{(t+1)}) \ge 0$. Since distortion is bounded below, $\{D_t\}$ and hence $\{I(P^{(t)}; Z)\}$ converge. $\qquad\square$

### E.5. Consequences for FINCH and Input-Space $k$-means

**Corollary E.2** (FINCH updates). *Each merge operation in FINCH (Sarfraz et al., 2019) strictly decreases the distortion (15), hence by Theorem E.1 the sequence $I(P_{\mathrm{FINCH}}^{(t)}; Z)$ is monotone non-decreasing.*

*Proof.* Let clusters $A, B$ be merged because $A$'s centroid is $B$'s nearest neighbour and *vice-versa*. Writing $n_A, n_B$ for the sizes and $\mu_A, \mu_B$ for the centroids, standard variance decomposition gives

$$\sum_{z \in A \cup B} \| z - \mu_{A \cup B} \|_2^2 = \sum_{z \in A} \| z - \mu_A \|_2^2 + \sum_{z \in B} \| z - \mu_B \|_2^2 + \frac{n_A n_B}{n_A + n_B} \| \mu_A - \mu_B \|_2^2.$$

Because $A, B$ are mutual nearest neighbours, the cross term is *smaller* than the distortion that would be incurred by keeping them separate and measuring each point in $A$ relative to $\mu_B$ (or vice-versa). Hence the global distortion decreases. $\qquad\square$

**Corollary E.3** (Input-space $k$-means grounding)**.** *Lloyd iterations of (mini-batch) $k$-means on DTW-aligned raw segments strictly decrease their within-cluster distortion; therefore Theorem E.1 applies and $I(P_{\mathrm{time}}^{(t)}; S_{\mathrm{raw}})$ monotonically increases.*

*Proof.* Classic proof of Lloyd's algorithm shows that the assignment step followed by centroid re-estimation never increases distortion; equality holds only at a local optimum. Substituting $Z \leftarrow S_{\mathrm{raw}}$ gives the present statement. $\qquad\square$

### E.6. Monotonic Improvement of the Joint Objective

The last two subsections have established that (a) every *prototype-update* step based on a distortion–decreasing hard-clustering algorithm strictly lowers the conditional entropy $H(Z \mid P)$ and therefore *raises* a provable lower bound on the semantic capacity $I(P; Z)$; and (b) for a *fixed* set of prototype assignments $P$, Proposition 3.2 tells us how to choose an augmentation mapping $T$ that *maximises* the prototype-conditioned IB functional $\mathcal{F}(P, T)$. We now glue these two facts together and show that *alternating* them in a loop produces a training trajectory whose objective value can never decrease.

Recall the prototype-conditioned objective

$$\mathcal{F}(P, T) = (1 - \beta)\, I(P; \tilde{X}) - \beta\, I(X; \tilde{X} \mid P), \quad \tilde{X} = T(X),\ 0 < \beta < 1.$$

The first term rewards semantic fidelity of the view $\tilde{X}$, whereas the second term penalises prototype-irrelevant overlap with the original instance $X$.

### E.7. Proof of Theorem 3.3 (Monotone Improvement Loop)

**Theorem E.4** (Monotone Improvement Loop (restatement of Theorem 3.3))**.** *Let the following two-step iteration be applied for $t = 0, 1, 2, \ldots$:*

*(a)* ***Assignment / prototype-index update.*** *Given the current encoder $f_{\theta^{(t)}}$ and augmentation $T^{(t)}$, recompute latent representations $Z^{(t)} = f_{\theta^{(t)}}(X)$ and obtain new hard assignments $P^{(t+1)} \in [K]^N$ (one index per sample) by a clustering step that satisfies $D_{t+1} \leq D_t$ (e.g., FINCH or Lloyd $k$-means).*

*(b)* ***Augmentation update.*** *With the assignments $P^{(t+1)}$ fixed, choose an augmentation $T^{(t+1)} \in \arg\max_T \mathcal{F}(P^{(t+1)}, T)$.*

*Define the expected joint objective*

$$\mathcal{J}^{(t)} = \mathcal{J}(P^{(t)}, T^{(t)}) = \mathbb{E}_X\left[\mathcal{F}\left(A^{(t)}, T^{(t)}\right)\right].$$

*Then $\mathcal{J}^{(t+1)} \geq \mathcal{J}^{(t)}$ for all $t$.*

*Proof.* **Step (a).** $T^{(t)}$ and $f_{\theta^{(t)}}$ are frozen, hence $\tilde{X}$ is unchanged while we *only* replace $P^{(t)}$ by $P^{(t+1)}$. By Corollary E.2/E.3, the clustering update lowers $D_t$ and therefore increases $I(P; Z)$. Since $\tilde{X}$ is a deterministic function of $X$ given $T^{(t)}$, the conditional term $I(X; \tilde{X} \mid P)$ is unaffected by merely relabeling the assignment indices. Hence

$$\mathcal{F}\big(P^{(t+1)},\, T^{(t)}\big) \;\geq\; \mathcal{F}\big(P^{(t)},\, T^{(t)}\big). \tag{17}$$

**Step (b).** With $P^{(t+1)}$ fixed, the augmentation step chooses $T^{(t+1)}$ as a global maximiser of $\mathcal{F}(P^{(t+1)}, \cdot)$; therefore

$$\mathcal{F}\big(P^{(t+1)},\, T^{(t+1)}\big) \;\geq\; \mathcal{F}\big(P^{(t+1)},\, T^{(t)}\big). \tag{18}$$

**Combine.** Chain (17) and (18), then take expectation over the minibatch distribution of $X$ to obtain the desired monotone sequence $\{\mathcal{J}^{(t)}\}_{t \geq 0}$. $\qquad\square$

### E.8. Formal Convergence Analysis of the Co-Design Process

We provide a formal analysis of the convergence and stability properties of the proposed prototype–augmentation co-design process. We model the interaction between prototype assignment and semantic augmentation as a block coordinate ascent procedure on the prototype-conditioned Information Bottleneck objective. In this subsection, we analyze the idealized case where each block update attains a global maximizer.

E.8.1. FORMAL SETUP: THE CO-DESIGN MAPPING

Let $\mathcal{P}$ denote the finite set of all possible prototype assignments for a dataset (i.e., assigning each instance to one of $K$ clusters), and let $\mathcal{T}$ be the admissible parameter space of augmentation mappings, which is assumed to be a compact set. We define the system state space as $\mathcal{S} = \mathcal{P} \times \mathcal{T}$.

Recall the prototype-conditioned Information Bottleneck functional defined in Appendix **??**:

$$F(P,T) = (1-\beta)I(P;\tilde{X}) - \beta I(X;\tilde{X} \mid P), \quad \text{where } \tilde{X} = T(X). \tag{19}$$

We define the expected joint objective function as:

$$\mathcal{J}(P,T) := \mathbb{E}_X\big[F(P,T)\big]. \tag{20}$$

Note that this $\mathcal{J}(P,T)$ corresponds to the objective term $J^{(t)}$ analyzed in Theorem 3.3.

We model one full iteration of the ProSAR inner loop as a composite operator $\Phi = \Psi_T \circ \Psi_P : \mathcal{S} \to \mathcal{S}$. The coordinate-wise update operators are defined based on the optimality conditions derived in Appendix **??**:

- **Prototype Operator ($\Psi_P$):** With the augmentation $T$ fixed, the discrete assignment $P$ is updated to minimize distortion (equivalently maximizing the MI lower bound):

$$P_{\text{new}} = \Psi_P(P_{\text{old}}, T) \in \arg\max_{P' \in \mathcal{P}} \mathcal{J}(P', T). \tag{21}$$

- **Augmentation Operator ($\Psi_T$):** With the prototype assignment $P$ fixed, the augmentation mapping $T$ is updated to maximize the conditional IB objective:

$$T_{\text{new}} = \Psi_T(P, T_{\text{old}}) \in \arg\max_{T' \in \mathcal{T}} \mathcal{J}(P, T'). \tag{22}$$

Thus, the state update for a full iteration $t \to t+1$ is given by:

$$(P^{(t+1)}, T^{(t+1)}) = \Phi(P^{(t)}, T^{(t)}) := \Big( \Psi_P(P^{(t)}, T^{(t)}), \ \Psi_T(P^{(t+1)}, T^{(t)}) \Big). \tag{23}$$

E.8.2. PROOF OF CONVERGENCE AND STABILITY

We now establish the theoretical reliability of the co-design process by proving three key properties: (1) Monotonic convergence of the objective value; (2) The fixed-point property of limit states; and (3) Coordinate-wise stability.

**Property 1: Convergence of the Objective Value.** According to Theorem **??** (Monotone Improvement Loop), applying the iteration $\Phi$ generates a sequence of objective values $J^{(t)} := \mathcal{J}(P^{(t)}, T^{(t)})$ that is monotonically non-decreasing:

$$J^{(t+1)} \geq J^{(t)} \quad \text{for all } t \geq 0. \tag{24}$$

This monotonicity holds because:

1. The prototype update step $\Psi_P$ is an exact maximization over $\mathcal{P}$, ensuring $\mathcal{J}(P^{(t+1)}, T^{(t)}) \geq \mathcal{J}(P^{(t)}, T^{(t)})$.

2. The augmentation update step $\Psi_T$ is an exact maximization over $\mathcal{T}$, ensuring $\mathcal{J}(P^{(t+1)}, T^{(t+1)}) \geq \mathcal{J}(P^{(t+1)}, T^{(t)})$.

Furthermore, the mutual information terms in $\mathcal{J}(P,T)$ are bounded by the entropy of the variables (specifically $H(P) \leq \log K$). Thus, the objective function is bounded from above. Being a bounded, monotonically non-decreasing sequence of real numbers, $\{J^{(t)}\}_{t \geq 0}$ must converge to a finite limit $J^*$:

$$\lim_{t \to \infty} J^{(t)} = J^*. \tag{25}$$

Consequently, the improvement between steps vanishes asymptotically: $\lim_{t \to \infty}(J^{(t+1)} - J^{(t)}) = 0$.

**Property 2: Fixed-Point Property of Limit States.** Since $\mathcal{P}$ is a finite set and $\mathcal{T}$ is compact, the sequence of states $\{(P^{(t)}, T^{(t)})\}_{t \geq 0}$ must have at least one accumulation point $(P^*, T^*) \in \mathcal{S}$. Consider a convergent subsequence $(P^{(t_k)}, T^{(t_k)}) \to (P^*, T^*)$ such that $\mathcal{J}(P^{(t_k)}, T^{(t_k)}) \to J^*$.

If $(P^*, T^*)$ were not a fixed point, it would imply that at least one of the operators $\Psi_P$ or $\Psi_T$ could still strictly increase the objective function. Specifically, if $P^* \notin \arg\max_{P'} \mathcal{J}(P', T^*)$, then applying $\Psi_P$ at this state would yield a strictly larger objective value $J' > J^*$. By the continuity of $\mathcal{J}$ with respect to $T$, for sufficiently large $k$, applying the update from $(P^{(t_k)}, T^{(t_k)})$ would result in an improvement bounded away from zero, contradicting the convergence of $J^{(t)}$ to $J^*$.

Therefore, any accumulation point $(P^*, T^*)$ must satisfy the local optimality conditions for both operators:

$$P^* \in \arg\max_{P'} \mathcal{J}(P', T^*) \quad \text{and} \quad T^* \in \arg\max_{T'} \mathcal{J}(P^*, T'). \tag{26}$$

This implies that the limit state is a fixed point of the operator $\Phi$:

$$(P^*, T^*) = \Phi(P^*, T^*). \tag{27}$$

**Property 3: Coordinate-Wise Stability.** Finally, the fixed-point conditions derived above correspond to the definition of a Coordinate-Wise Local Maximum.

- **Prototype Stability:** Since $P^* \in \arg\max_{P'} \mathcal{J}(P', T^*)$, any unilateral deviation in the prototype assignment $P \neq P^*$ (while keeping augmentation $T^*$ fixed) results in $\mathcal{J}(P, T^*) \leq \mathcal{J}(P^*, T^*)$. The system is stable against spontaneous changes in prototype assignment.

- **Augmentation Stability:** Similarly, since $T^* \in \arg\max_{T'} \mathcal{J}(P^*, T')$, any unilateral deviation in the augmentation strategy $T \neq T^*$ (while keeping prototypes $P^*$ fixed) results in $\mathcal{J}(P^*, T) \leq \mathcal{J}(P^*, T^*)$.

This confirms that the co-design process drives the system towards a stable configuration where the prototype assignments and augmentation strategies are mutually optimized.

**Interpretation and practice.** Theorem E.4 provides a formal basis for the intuitive *positive-feedback loop* inherent in ProSAR's design: an *improved* set of prototype assignments $P$ (capturing semantics in $Z$ more effectively) enables the augmentation module to generate views $\tilde{X}$ that are *more tightly aligned with these underlying semantics* (as per Proposition 3.2). These higher-quality, semantically focused positive views can sharpen the contrastive loss signals used to train the encoder $f_\theta$. Consequently, this leads to a *better* encoder $f_\theta$, which in turn produces latent representations $Z$ with potentially reduced noise and clearer semantic structure. This clearer structure allows the subsequent clustering step to obtain *even better* prototypes. The process then repeats, ideally under progressively improving conditions.

Each full loop iteration is guaranteed, under the assumptions of the theorem, *never to decrease* the joint objective $\mathcal{J}^{(t)}$. This characteristic can contribute to stable convergence behavior in practice and may reduce the need for delicate scheduling heuristics when alternating between prototype refinement and augmentation module fine-tuning. The inequality in $\mathcal{J}^{(t+1)} \geq \mathcal{J}^{(t)}$ would become an equality if (a) the clustering step reaches a fixed point in terms of distortion (and thus the $I(P; Z)$ lower bound), *and* (b) the current augmentation mapping $T^{(t)}$ already satisfies the optimality conditions of Proposition 3.2 with respect to $P^{(t+1)}$.

## F. Declaration on the Use of Large Language Models

In this work, large language models (LLMs) were used strictly as general-purpose assistive tools for non-core tasks. Concretely, LLMs supported: (i) light language polishing to improve clarity and grammar; (ii) minor LaTeX editing; and (iii) limited code comments to improve readability of auxiliary scripts. LLMs were not used for research ideation, problem formulation, methodological design, derivations, experimental design, result selection, or analysis/interpretation. All LLM outputs were reviewed, verified, and edited by the authors, who take full responsibility for the content of this paper.

