# OpenReview forum: "ProSAR: Prototype-Guided Semantic Augmentation and Refinement for Time Series Contrastive Learning"
_ICML.cc/2026/Conference — ICML 2026 regular_

### Official Review · Reviewer_QRfP · 2026-03-10

**Soundness:** 3
**Presentation:** 2
**Significance:** 3
**Originality:** 3
**Overall Recommendation:** 5
**Confidence:** 4

**Summary:**

The paper addresses an important problem in time-series self-supervised learning: standard augmentation methods may damage semantically meaningful temporal structure and are often difficult to interpret. To address this, the paper proposes **ProSAR, a prototype-guided self-supervised framework that uses an information-theoretic perspective** to design semantic-preserving augmentations. The core idea is to use explicit prototype alignment to identify semantically relevant segments, apply different transformations to semantic and non-semantic parts, and iteratively refine the prototypes during training. The method is evaluated on forecasting and classification tasks and achieves improved performance over several competitive baselines.

**Compliance With Llm Reviewing Policy:**

Affirmed.

**Final Justification:**

**Thank you to the authors for the detailed rebuttal and the additional experiments.**

My main concerns have been substantially addressed, and I therefore raise my score from 4 to 5.

That said, as the authors also acknowledged, the revised version should include a more comprehensive set of baselines and settings for transfer learning and semi-supervised classification. Furthermore, to enable a more accurate performance comparison beyond the reported results, experiments should ideally be conducted under the same physical environment and experimental setup.

**Key Questions For Authors:**

1. Can you provide stronger evidence that the full closed-loop feedback mechanism is a primary source of the gains?
2. Can you clarify the computational overhead of the method, especially for MI estimation and clustering, and compare it with relevant baselines?
3. Can you provide broader experimental validation, such as on more datasets or in settings like transfer learning or semi-supervised learning?
4. Can you report standard deviations and present results for closely related baselines such as MHCCL and AimTS more prominently, ideally beyond only averages?

**Limitations:**

yes

**Strengths And Weaknesses:**

## Soundness
- The paper provides a theoretically reasonable and logically connected information-theoretic framing for both augmentation design and prototype refinement. While the theory is not directly optimized end-to-end, it offers a meaningful justification for the proposed method.
- The paper emphasizes the closed-loop co-design between prototype refinement and augmentation as a central contribution. However, the ablation results suggest that removing decoder feedback leads to a relatively modest degradation, whereas removing DTW-based segmentation, STFT-based transformation, or clustering causes larger drops. This raises some doubt about whether the full feedback loop is truly the primary source of improvement.
- The empirical study is reasonably designed, but its scope is still somewhat limited. Additional evaluations on more datasets and under broader settings, such as transfer learning or semi-supervised learning, would strengthen the claims.
- The computational overhead is not sufficiently analyzed. In particular, the cost of MI estimation and clustering should be discussed more clearly, and comparisons of training time or augmentation overhead against relevant baselines would improve the evaluation.

## Presentation
- The paper is generally clear, well organized, and easy to follow.
- Reporting standard deviations for the main results would improve the clarity of the evaluation. In addition, results for closely related methods such as MHCCL and AimTS should be presented more prominently, for example through per-dataset comparisons rather than only averages.
- Some implementation details remain unclear for reproducibility, especially the decay schedule of the ISA fusion weight $\alpha$ used for time-prototype updates.

## Significance
- The paper addresses an important problem in time-series self-supervised learning, where existing augmentation methods may destroy semantically important temporal dependencies and limit interpretability and controllability.
- The proposed information-theoretic framework and the closed-loop co-design perspective between prototype refinement and augmentation are potentially useful beyond this particular method, and may inspire future work on semantic-aware augmentation for time-series representation learning.

## Originality
- The paper presents a novel combination of ideas by proposing an information-theoretic framework that leverages explicit prototype alignment to guide semantic augmentations, while also establishing a feedback loop between augmentation, contrastive learning, and prototype updates.

---

> ### Author Rebuttal · Authors · 2026-03-31
>
> We would like to thank the reviewer for the comments. Due to the response-length limit, we summarize only the key points here, with supporting figures/tables provided at https://anonymous.4open.science/r/ICML26-B2FC/Results.pdf.
>
> ### **Comment 1: Full closed-loop feedback**
>
> **Response:**
> Considering that the original ablation might be insufficient, we add a targeted feedback-path ablation in **Table A7**.
>
> On univariate forecasting, the full model achieves **0.151/0.250** (MSE/MAE). Removing decoder feedback (**w/o Dec.**) gives **0.164/0.261**; removing input-space anchoring (**w/o ISA**) gives **0.159/0.257**; removing both (**w/o Dec.\&ISA**) gives **0.180/0.274**. For reference, disabling clustering (**w/o Clust.**) gives **0.175/0.270**.
>
> These results show that both the decoder feedback and ISA contribute to prototype refinement. The moderate drop of **w/o Dec.** is because ISA still updates time-domain prototypes. Since **w/o ISA** outperforms **w/o Dec.**, decoder-based latent-to-time feedback appears more important. We'd like to further clarify our inented claim: the closed-loop feedback is a key innovation of ProSAR and an important source of the gains, while the final improvement comes from the joint effect of multiple modules rather than the feedback alone.
>
> ### **Comment 2: Computational overhead**
>
> **Response:**
> We add runtime analysis and comparison with baselines in **Table A4** and **Fig. A1**.
>
> At the module level in **Table A4**, the average runtime is **848.0 ms/epoch**, of which **loss+backward** accounts for **690.972 ms (81.48%)**. In contrast, **DTW alignment+matching** takes **24.304 ms (2.87%)**, **FINCH clustering** takes **72.452 ms (8.49%)**, **MI refinement** and **frequency augmentation** take **19.612 ms (2.31%)** and **12.388 ms (1.46%)**.
>
> In Fig. A1, over 20 epochs, average training time per epoch is **0.848 s** for **ProSAR**, versus **0.583 s** for TimesURL [D1], **1.296 s** for FreRA [D2], and **1.552 s** for AutoTCL [D3]. Thus, ProSAR adds moderate overhead and remains faster than several strong baselines.
>
> Further, these extra computations are used only in self-supervised pre-training. Under the frozen-encoder downstream protocol, no DTW-based augmentation, clustering, or prototype refinement is used at inference time, so deployment-time complexity is unchanged.
>
> ### **Comment 3: Standard deviation and closely related baselines**
>
> **Response:**
> We add **mean±std** results and comparisons with AimTS and MHCCL in **Tables A8** and **A9**.
>
> Across five seeds, ProSAR consistently outperforms these two baselines with low variance. In univariate forecasting, ProSAR achieves **0.151±0.001/0.250±0.001** (MSE/MAE), vs. **0.171±0.001/0.284±0.002** for AimTS and **0.176±0.001/0.274±0.002** for MHCCL. In classification, ProSAR achieves **0.765±0.003**,vs. **0.714±0.005** for AimTS and **0.706±0.002** for MHCCL.
>
>
>
>
> ### **Comment 4: ISA fusion**
>
> **Response:**
> We will state this more explicitly in the revised manuscript. The current mechanism is **matching+weighted fusion+EMA smoothing**. Time-domain candidates come from empirical centroids $\{c_j\}$ from input-space anchoring and decoded prototypes $\{\hat p_j^x\}$ from the decoder. For each active time-domain prototype $p_i^x$, correspondence is established by nearest matching:
> $$
> n(i)=\arg\min_j \left\|p_i^x-c_j\right\|_2,\qquad
> m(i)=\arg\min_j \left\|p_i^x-\hat p_j^x\right\|_2 .
> $$
>
> Then, the fused candidate is
> $$
> \tilde p_i^x=\alpha_x\, c_{n(i)}+(1-\alpha_x)\,\hat p_{m(i)}^x ,
> $$
> and the EMA update is
> $$
> p_i^x \leftarrow \alpha_z\, p_i^x + (1-\alpha_z)\,\tilde p_i^x ,
> $$
> where $\alpha_x$ is the fusion weight and $\alpha_z$ is the EMA momentum.
> In the current implementation, $\alpha_x$ follows a linear decay schedule: it is set larger in early training to rely more on ISA, and gradually decreases later so that refinement depends more on decoded prototype feedback.
>
> ### **Comment 5: Broader validation**
>
> **Response:**
> We extend the evaluation in **Table A10** to **ETTm2** and **Exchange**, and compare ProSAR with AutoTCL, FreRA, and TimesURL.
>
> On **ETTm2**, ProSAR achieves **0.563/0.490** (MSE/MAE), vs. **0.568/0.497** for AutoTCL, **0.578/0.510** for FreRA, and **0.652/0.543** for TimesURL. On **Exchange**, ProSAR achieves **0.707/0.568**, vs. **0.785/0.608** for AutoTCL, **0.795/0.621** for FreRA, and **0.835/0.622** for TimesURL. ProSAR also achieves the best on nearly all forecasting horizons in both datasets.
>
> Due to the rebuttal-time budget, we focus on validation over additional datasets, while transfer-learning or semi-supervised settings will be our future extensions.
>
>
>
> [D1] TimesURL: Self-supervised contrastive learning for universal time series representation learning, AAAI 2024.
>
> [D2] FreRA: A frequency-refined augmentation for contrastive learning on time series classification, ACM SIGKDD 2025.
>
> [D3] Parametric augmentation for time series contrastive learning, ICLR 2024.

---

> > ### Author Rebuttal · Reviewer_QRfP · 2026-04-02
> >
> > **Thank you for the authors’ sincere response and the additional experiments.**
> >
> > My concerns regarding Q1, Q2, and Q4 have been adequately addressed.
> >
> > Regarding Q3, I understand that, due to the limited rebuttal-time budget, the authors focused on further validation on additional datasets, particularly univariate datasets, while leaving transfer learning and semi-supervised settings for future work. While this is understandable, I still find this somewhat unfortunate, as transfer learning and semi-supervised settings are particularly important in the SSL context.
> >
> > At this stage, I will maintain my current score of weak accept. However, I will increase my assessment of soundness and confidence based on the authors’ response.
> >
> > If the authors are able to provide even a brief experimental validation on transfer learning or semi-supervised settings, I would be willing to further increase my final score.

---

> > > ### Author Response · Authors · 2026-04-07
> > >
> > > Thank you for the helpful follow-up comment and for highlighting this important point. We agree that a broader validation in **transfer learning** and **semi-supervised settings** is especially valuable for assessing the SSL methods. To address this concern, we have additionally conducted experiments in both of the two settings, following the implementation details and evaluation setup of the corresponding prior works.
> > >
> > > #### (1) Transfer learning
> > > We first evaluate **ProSAR** under the transfer-learning setting used in **FreRA [D2]**. Specifically, on the SHAR dataset, the encoder is pre-trained on a set of source domains and then directly transferred to an unseen target domain for evaluation. Following FreRA, the number of source domains (No.SD) is 3 and 19, and TD denotes the target-domain index. In **Table A11**, the baseline results are taken from the original **FreRA** paper, and **ProSAR** is evaluated under the same setting for a fair comparison. As shown in this table, **ProSAR** achieves the best performance in **7 out of 8** transfer settings and ranks **second** in the remaining one, which can demonstrate that the learned representations transfer effectively across domains.
> > >
> > > **Table A11.** Classification performance under the transfer-learning setting on the **SHAR** dataset across different numbers of source domains. The best result is highlighted in **bold**, and the second-best result is in *italics*.
> > > | No. SD | TD | ProSAR | FreRA | AutoTCL | TS2Vec | TS-TCC | SoftCLT (TS-TCC-based) |
> > > |---|---:|---:|---:|---:|---:|---:|---:|
> > > | 3  | 1 | **0.604** | *0.602* | 0.464 | 0.430 | 0.495 | 0.505 |
> > > | 3  | 2 | **0.516** | *0.467* | 0.278 | 0.317 | 0.410 | 0.407 |
> > > | 3  | 3 | **0.684** | *0.665* | 0.414 | 0.523 | 0.464 | 0.530 |
> > > | 3  | 5 | **0.406** | *0.366* | 0.245 | 0.050 | 0.362 | 0.339 |
> > > | 19 | 1 | **0.654** | *0.628* | 0.497 | 0.568 | 0.578 | 0.581 |
> > > | 19 | 2 | **0.664** | *0.652* | 0.372 | 0.640 | 0.647 | 0.581 |
> > > | 19 | 3 | **0.697** | *0.691* | 0.408 | 0.502 | 0.592 | 0.559 |
> > > | 19 | 5 | *0.695* | **0.698** | 0.430 | 0.658 | 0.612 | 0.567 |
> > > #### (2) Semi-supervised classification
> > >
> > > We further evaluate **ProSAR** under the **1% labeled-data setting** used in **SoftCLT [D4]**. Following SoftCLT, we consider the semi-supervised classification setting of self-supervised learning with unlabeled data,  followed by a supervised fine-tuning with 1% labeled data. In **Table A12**, the results of **TS2Vec**, **SoftCLT (TS2Vec-based)**, **TS-TCC**, and **SoftCLT (TS-TCC-based)** are taken from the original **SoftCLT** paper, while the results of **ProSAR**, **FreRA**, and **AutoTCL** are obtained from our additional experiments under the same setting. As shown in this table, **ProSAR** achieves the best **ACC** on **7 out of 8** datasets and consistently outperforms the previous semantic-aware baselines **FreRA** and **AutoTCL** on all the eight datasets. These results further demonstrate that **ProSAR** improves the label efficiency in the low-label SSL scenarios.
> > >
> > > **Table A12.** Semi-supervised classification results under the **1% labeled-data setting**. Results are reported as **ACC/MF1**.
> > >
> > > | Dataset | ProSAR | FreRA | AutoTCL | TS2Vec | SoftCLT (TS2Vec-based) | TS-TCC | SoftCLT (TS-TCC-based) |
> > > |---|---:|---:|---:|---:|---:|---:|---:|
> > > | HAR | **0.915 / 0.915** | 0.901 / 0.903 | 0.783 / 0.774 | 0.886 / 0.885 | *0.910* / *0.910* | 0.705 / 0.695 | 0.829 / 0.828 |
> > > | Epilepsy | **0.964 / 0.963** | 0.957 / 0.955 | 0.928 / 0.924 | 0.958 / 0.934 | *0.963* / 0.941 | 0.912 / 0.892 | 0.956 / *0.956* |
> > > | Wafer | **0.978** / *0.941* | 0.964 / 0.900 | 0.936 / 0.784 | 0.679 / 0.561 | 0.953 / 0.881 | 0.932 / 0.767 | *0.965* / **0.965** |
> > > | FordA | **0.896 / 0.895** | *0.872* / *0.872* | 0.853 / 0.853 | 0.864 / 0.864 | 0.871 / 0.871 | 0.806 / 0.800 | 0.815 / 0.812 |
> > > | FordB | *0.782* / *0.782* | 0.774 / 0.774 | 0.751 / 0.751 | 0.654 / 0.654 | 0.679 / 0.679 | **0.786 / 0.786** | 0.748 / 0.748 |
> > > | POC | **0.720 / 0.671** | *0.682* / 0.633 | 0.675 / 0.626 | 0.631 / 0.628 | 0.636 / 0.628 | 0.638 / 0.481 | 0.654 / *0.646* |
> > > | StarLightCurves | **0.914 / 0.863** | *0.908* / *0.853* | 0.893 / 0.804 | 0.829 / 0.606 | 0.856 / 0.629 | 0.860 / 0.792 | 0.860 / 0.793 |
> > > | ElectricDevices | **0.651** / **0.632** | 0.633 / 0.553 | 0.629 / 0.551 | 0.576 / 0.486 | 0.620 / 0.530 | 0.636 / *0.564* | *0.646* / **0.632** |
> > >
> > >
> > >
> > > Overall, due to the limited time of rebuttal period, we might not be able to cover all the possible baselines and settings. We sincerely hoped that these additional results on transfer learning and semi-supervised classification could provide a meaningful response to your concern.
> > >
> > >
> > > [D2] FreRA: A frequency-refined augmentation for contrastive learning on time series classification, ACM SIGKDD 2025.
> > >
> > > [D4] Soft Contrastive Learning for Time Series, ICLR 2024.

---

### Official Review · Reviewer_NQgT · 2026-03-11

**Soundness:** 3
**Presentation:** 3
**Significance:** 3
**Originality:** 3
**Overall Recommendation:** 4
**Confidence:** 3

**Summary:**

This paper proposes ProSAR, a contrastive learning framework for time series data augmentation. Rather than applying static heuristics, it uses prototype-guided augmentation policies: semantically meaningful segments are identified via soft-DTW alignment with time-domain prototypes, preserved through STFT phase compensation, while non-semantic regions are heavily perturbed. A dual-prototype feedback loop iteratively refines latent and time-domain anchors. The method is grounded in a prototype-conditioned information bottleneck objective and evaluated on standard forecasting (ETTh, Electricity, Weather) and classification (UEA) benchmarks, outperforming contrastive learning baselines such as AutoTCL and FreRA.

**Compliance With Llm Reviewing Policy:**

Affirmed.

**Final Justification:**

Thank the authors for their responses.

Comment 1: The clarification that flattening is used only as an alignment surrogate, with representation learning operating on the full multivariate input is satisfactory (+ empirical check (0.151 → 0.153 MSE) further confirms this is not a source of systematic harm). Comment 2: This addresses my concern. Comment 3: The visualizations in Appendix D.1 provide reasonable qualitative support. Comment 4: The paired t-tests with five-seed results and reported p-values fully address this concern.

One remaining suggestion: the main paper would benefit from tighter writing, as verbosity was noted as a weakness that was not directly addressed in the rebuttal. I maintain my evaluation.

**Key Questions For Authors:**

1. How are multivariate inputs handled in the soft-DTW alignment and prototype matching stages? Does channel flattening not discard inter-channel semantic structure, and if so, how is this reconciled with the framework's stated objectives?

2. Were statistical significance tests conducted on the reported improvements? If not, how can the authors rule out that performance gains fall within the range of random variation?

3. What formal or empirical evidence supports the claim that prototypes are more interpretable than alternative representation approaches?

4. How sensitive is the framework to the initialization strategy for latent and time-domain prototypes?

**Limitations:**

Yes

**Strengths And Weaknesses:**

** Strengths

- The paper is overall clear. The Prototype-Conditioned Information Bottleneck mathematically justifies the dual augmentation strategy of preserving xs while corrupting xn, and the pipeline is reproducible(thank you for submitting the code in the supplementary materials).
- Designing semantics-preserving augmentations for time series contrastive learning is a genuine and open challenge. The prototype-as-anchor perspective is compelling and has potential relevance beyond contrastive learning, e.g., in counterfactual explanation generation.
- Table 9 is comprehensive; each component is validated independently, providing convincing evidence that contributions are additive rather than coincidental.
- The CUDA soft-DTW implementation adds only ~1 ms per iteration (~2.5% epoch overhead) with zero inference cost. The runtime analysis in Appendix C.9 is welcome and practically important.

**Weaknesses

- Based on the code (cf supp. mat.), multivariate inputs appear to be handled by flattening channels before soft-DTW alignment and prototype matching. This design choice is never mentioned or justified in the paper. Flattening discards inter-channel dependencies, which are precisely the kind of semantic structure the framework aims to preserve. The authors must clarify and justify this decision.
- Section 3 introduces the information bottleneck objective with several propositions and a convergence theorem, but the connection between these theoretical results and the concrete implementation (DTW-based segment identification and frequency-domain perturbations) is never made explicit. Readers are left to infer how the formalism translates into design choices.
- The claim that prototypes are more interpretable than alternative representations is stated but never formally or empirically substantiated, weakening what is presented as a key advantage of the method.
- Average rankings are reported, but the absence of significance tests makes it difficult to determine whether observed improvements exceed the range of random variation. This concern is amplified by the relatively small gains in some settings (e.g., classification accuracy improves by only ~0.01 over FreRA on average across UEA datasets).
- The main paper is overly verbose, making it hard to follow over long stretches.

---

> ### Author Rebuttal · Authors · 2026-03-31
>
> We would like to thank the reviewer for the comments. Due to the response-length limit, we summarize only the key points here, with supporting figures/tables provided at https://anonymous.4open.science/r/ICML26-B2FC/Results.pdf.
>
> ### **Comment 1: Multivariate flattening and its justification**
>
> **Response:**
> In our current implementation, channel flattening is used only in the soft-DTW alignment and prototype-matching stage, as a surrogate for temporal correspondence estimation. It is not used throughout the full framework. After alignment, the learned masks are applied to the original multivariate input, and then the encoder, contrastive learning, and downstream representation learning all operate on the original multivariate series. Thus, inter-channel dependencies are still modeled in the actual representation-learning stage.
>
> We also tested a non-flattened variant on the forecasting task and found that its average MSE slightly degrades from **0.151** to **0.153**. This suggests that flattening is not the source of the performance gain (or degradation). Rather, it is used only as a limited alignment surrogate under a controlled complexity budget.
>
> ### **Comment 2: Theory to design**
>
> **Response:**
> We will state the theory-to-design mapping more explicitly in the revised manuscript. Our starting point is the prototype-conditioned information bottleneck objective
> $$
> \max_T I(P;\tilde X)-\beta I(X;\tilde X),
> $$
> where $T$ denotes the prototype-guided augmentation policy. Here, $I(P;\tilde X)$ encourages $\tilde X$ to preserve information consistent with prototype $P$, while $\beta I(X;\tilde X)$ prevents the augmentation from trivially preserving the full input.
>
> This maps directly to the implementation. DTW-based alignment identifies prototype-consistent regions and constructs a mask $M_x$, partitioning the input into
> $$
> x_S=x\odot M_x,\qquad x_N=x\odot(1-M_x).
> $$
> The preservation term $I(P;\tilde X)$ is implemented by transforming $x_S$ more carefully so that prototype-consistent structure is retained. The compression term $\beta I(X;\tilde X)$ is implemented by applying stronger perturbation to $x_N$. Since the objective is conditioned on the current prototype assignment $P$, ProSAR further introduces a refinement loop: latent prototypes are updated by clustering, and decoded feedback refines time-domain prototypes so that the next-round augmentation remains aligned with the learned semantic structure.
>
> ### **Comment 3: Interpretability of  prototypes**
>
> **Response:**
> We tried to empirically show this in Appendix D.1, where the learned time-domain prototypes evolve into waveforms with clear periodic structure. We also showed that DTW alignment paths make the matching between input segments and prototypes directly visible. These results support the claim that prototypes provide a more inspectable semantic anchor than the implicit masking.
>
> ### **Comment 4: Average rankings without significance test**
>
> **Response:**
> We agree that average rankings alone are insufficient, and thus add statistical significance tests and report mean±std in **Tables A5** and **A6**.
>
> Using newly added five-seed results, we conduct **paired t-tests** between ProSAR and the closest baselines. For **univariate forecasting**, ProSAR is significantly better than **AutoTCL [C1]**, **FreRA [C2]**, and **TimesURL [C3]** on both MSE and MAE, with p-values **7.07e-05/0.0040**, **7.34e-04/8.92e-05** and **2.13e-04/ 2.32e-05**. For **classification**, ProSAR is also significantly better than AutoTCL, FreRA, and TimesURL, with p-values **4.34e-04**, **0.0020**, and **2.83e-04**, respectively.
>
> ### **Comment 5: Sensitivity to prototype initialization**
>
> **Response:**
> In the original experiments, all reported results use **random prototype initialization**, and we observed stable convergence without noticeable instability.
>
>
> We further compare the **full random initialization**, **sampled subsequences** (initializing prototypes from randomly sampled training segments), and **centroid warm-start** (initializing prototypes from cluster centroids of training segments). The final performance is very similar: **0.151/0.250** (MSE/MAE) for full random initialization, **0.152/0.251** for sampled subsequences, and **0.151/0.250** for centroid warm-start. This indicates that ProSAR is not strongly sensitive to initialization and does not rely on a specialized initialization scheme.
>
> In addition, the two data-dependent strategies do yield a **slightly faster early-stage convergence**, suggesting that initialization mainly affects the optimization speed rather than final performance.
>
> [C1] Parametric augmentation for time series contrastive learning, ICLR 2024.
>
> [C2] FreRA: a frequency-refined augmentation for contrastive learning on time series classification, ACM SIGKDD 2025.
>
> [C3] TimesURL: Self-supervised contrastive learning for universal time series representation learning, AAAI 2024.

---

> > ### Author Rebuttal · Reviewer_NQgT · 2026-04-02
> >
> > Thank the authors for their responses.
> >
> > Comment 1: The clarification that flattening is used only as an alignment surrogate, with representation learning operating on the full multivariate input is satisfactory (+ empirical check (0.151 → 0.153 MSE) further confirms this is not a source of systematic harm).
> > Comment 2:  This addresses my concern.
> > Comment 3: The visualizations in Appendix D.1 provide reasonable qualitative support.
> > Comment 4: The paired t-tests with five-seed results and reported p-values fully address this concern.
> >
> > One remaining suggestion: the main paper would benefit from tighter writing, as verbosity was noted as a weakness that was not directly addressed in the rebuttal. I maintain my evaluation.

---

> > > ### Author Response · Authors · 2026-04-07
> > >
> > > Thank you very much for your careful reading, thoughtful feedback, and positive follow-up. We are very glad that our rebuttal has addressed your main concerns.
> > >
> > > We also sincerely appreciate your constructive suggestions, especially regarding the justification of multivariate alignment, the theory-to-design connection, the interpretability discussion, the statistical significance analysis, and the sensitivity to prototype initialization. In addition, we agree with your final suggestion that the main paper would benefit from a tighter writing. In the revised manuscript, we will further incorporate these helpful comments and improve the presentation accordingly.

---

### Official Review · Reviewer_Mmrv · 2026-03-12

**Soundness:** 2
**Presentation:** 3
**Significance:** 3
**Originality:** 2
**Overall Recommendation:** 4
**Confidence:** 5

**Summary:**

This paper introduces ProSAR, which uses learnable prototypes to guide segmentation, augmentation, and refinement in time series contrastive learning. The idea is to preserve prototype-related information while removing noise. However, ProSAR lacks sufficient detail and experimental validation.

**Compliance With Llm Reviewing Policy:**

Affirmed.

**Final Justification:**

Weak accept

**Key Questions For Authors:**

Please refer to the weaknesses.

**Limitations:**

yes

**Strengths And Weaknesses:**

strength

1. They tackle a key issue in time series contrastive learning: generic augmentations like jittering and scaling often distort temporal semantics. The authors aim to make augmentations "semantic-aware," which is a sensible direction.
2. They integrate prototype-guided segmentation, targeted augmentation, and prototype refinement into a closed loop, resulting in a well-structured framework.
3. In the main table, ProSAR achieves decent average results on both forecasting and classification tasks.

weakness

1. The binary division between "semantic" and "non-semantic" is too arbitrary. The so-called "non-semantic" parts in time series are not necessarily irrelevant to the task. The authors do not demonstrate that this segmentation bias does not harm long-term dependencies and global patterns.
2. The authors emphasize that latent prototypes guide time-domain prototypes in reverse through the decoder, but only Section 3.2.2 provides a qualitative description of the latent-to-time-domain decoding consistency, and the main text does not specify the feedback optimization loss.
3. How is the number of prototypes selected? What if multiple prototypes are close to each other?
4. Section 3.2.2 states that "The final active time-domain prototypes are obtained by a weighted fusion of the input-space anchoring centroids and the decoder output," but it does not specify the fusion formula or how the one-to-one correspondence is achieved.
5. Although Appendix C.9 provides a theoretical analysis of efficiency, it would be better to include actual training time curves, as well as the impact of DTW and FINCH clustering, which are not lightweight modules themselves.

---

> ### Author Rebuttal · Authors · 2026-03-31
>
> We would like to thank the reviewer for the comments. Due to the response-length limit, we summarize only the key points here, with supporting figures/tables provided at https://anonymous.4open.science/r/ICML26-B2FC/Results.pdf.
>
> ### **Comment 1: Segmentation bias**
>
> **Response:**
> We agree that the non-semantic regions are not absolutely irrelevant to tasks. In our work, the semantic/non-semantic split is determined upon their alignment with the prototype under the current augmentation mechanism. Thus, more precisely, the *non-semantic* part is considered weakly prototype-inconsistent.
>
> To examine if this partition introduces a harmful segmentation bias, we further evaluate two alternative treatments to the weakly aligned regions (i.e., *non-semantic* part): a more aggressive perturbation (i.e., adding a larger noise) and a more conservative treatment (i.e., soft probabilistic selection of semantic/non-semantic parts). As shown in **Table A3**, the full model achieves on average **0.151/0.250** (MSE/MAE), compared to **0.153/0.252** under the larger noise and **0.156/0.255** under the soft selection. Overall, the performance degradation is relatively small, suggesting that such a partition does not essentially damage the global patterns or long-range dependencies.
>
> ### **Comment 2: Decoder feedback and optimization mechanism**
>
> **Response:**
> To address concern on the feedback optimization loss, we would like to further clarify the proposed two-stage training mechanism. First, a brief warmup stage jointly optimizes the encoder and decoder using a reconstruction loss, implemented as mean squared error (MSE), to establish an initial latent-to-time prototype mapping. Second, in the main training stage, the decoder is fixed and the encoder is trained with the contrastive loss. Though the decoder is frozen at this stage, its output is still refreshed because the latent prototypes are updated through clustering.
>
>
> ### **Comment 3: Prototype selection**
>
> **Response:**
> In our current implementation, the numbers of latent and time-domain prototypes, $K$ and $K_t$, are dataset-dependent hyperparameters selected by validation, with default settings $K=32$ and $K_t=32$. We furthe adopt EMA smoothing to reduce sensitivity to initialization.
>
> We additionally analyze sensitivity to the prototype count. On the univariate forecasting benchmark, increasing the number of prototypes from **16 to 32** improves the average MSE/MAE from **0.153/0.252** to **0.151/0.250**, while further increasing it to **64** keeps performance unchanged. This indicates that our method is **stable within a reasonable range**, rather than relying on a highly specific prototype number.
>
> When multiple prototypes are close to each other, the current mechanism does not perform hard replacement. Instead, prototypes are first updated through prototype matching and then smoothed by EMA, such that transient similarity or partial redundancy does not directly cause instability.
>
>
>
> ### **Comment 4: Fusion formula and one-to-one correspondence**
>
> **Response:**
> We will state this part more explicitly in the revised manuscirpt. The current mechanism follows a **matching+weighted fusion+EMA smoothing** procedure.
>
> Time-domain candidates come from the empirical centroids $\{c_j\}$ from the input-space anchoring and decoded prototypes $\{\hat p_j^x\}$ from the decoder. For each active time-domain prototype $p_i^x$, the correspondence is established by nearest-distance matching:
>
> $$
> n(i)=\arg\min_j \|p_i^x-c_j\|_2,\qquad
> m(i)=\arg\min_j \|p_i^x-\hat p_j^x\|_2.
> $$
>
> Then, the fused candidate is
>
> $$
> \tilde p_i^x=\alpha_x\, c_{n(i)}+(1-\alpha_x)\,\hat p_{m(i)}^x,
> $$
>
> followed by an EMA update:
>
> $$
> p_i^x \leftarrow \alpha_z\, p_i^x + (1-\alpha_z)\,\tilde p_i^x.
> $$
>
>
>
> ### **Comment 5: Runtime/overhead analysis**
>
> **Response:**
> To show the actual training time and impact of DTW and FINCH clustering, we add both **per-epoch training-time curves** and a **module-level runtime breakdown** in **Fig. A1** and **Table A4**.
>
> In Fig. A1, across 20 epochs, ProSAR averages **0.848 s/epoch**, compared to **0.583 s/epoch** for TimesURL [B1], **1.552 s/epoch** for AutoTCL [B2], and **1.296 s/epoch** for FreRA [B3]. In **Table A4**, the average runtime is **848.0 ms/epoch**, of which **loss + backward** accounts for **690.972 ms (81.48%)**. In contrast, **DTW alignment + matching** takes **24.304 ms (2.87%)**, **FINCH clustering** takes **72.452 ms (8.49%)**, **MI refinement** and **frequency augmentation** take **19.612 ms (2.31%)** and **12.388 ms (1.46%)**. Thus, main overhead comes from the backbone training, rather than DTW and FINCH clustering.
>
> [B1] TimesURL: Self-supervised contrastive learning for universal time series representation learning, AAAI 2024.
>
> [B2] Parametric augmentation for time series contrastive learning, ICLR 2024.
>
> [B3] FreRA: a frequency-refined augmentation for contrastive learning on time series classification, ACM SIGKDD 2025.

---

> > ### Author Rebuttal · Reviewer_Mmrv · 2026-04-05
> >
> > Thank you for your rebuttal, I will raise my score to weak accept

---

> > > ### Author Response · Authors · 2026-04-07
> > >
> > > Thank you very much for your careful reading, constructive feedback, and positive follow-up. We are glad that our rebuttal has addressed your concerns.
> > >
> > > We also sincerely appreciate your helpful comments on the segmentation bias, decoder feedback, prototype selection, fusion details, and runtime analysis. These suggestions have been very valuable for improving both the clarity and completeness of the paper. In the revised manuscript, we will further incorporate these comments by making the relevant implementation details and empirical analyses more explicit in the main text.

---

### Official Review · Reviewer_rXmg · 2026-03-16

**Soundness:** 3
**Presentation:** 3
**Significance:** 3
**Originality:** 3
**Overall Recommendation:** 4
**Confidence:** 4

**Summary:**

Summary: This paper proposes a feedback-aware data augmentation method for multi-variable time series learning. In particular, it involves a prototype-refinement process in time domain that iteratively enhance the augmentation strategy along the contrastive learning process. Extensive experiments show the superiority of this method compared to other techniques.

**Compliance With Llm Reviewing Policy:**

Affirmed.

**Final Justification:**

Thanks the authors for a detailed rebuttal that resolves my concern. And I will keep my original positive score.

**Key Questions For Authors:**

1. What may be different requirement for data augmentation and contrastive learning for different downstream tasks, e.g., forecasting and classification?
2. What is the scalability for this method in terms of data size in contrastive learning?

**Limitations:**

Yes

**Strengths And Weaknesses:**

1. Soundness: This submission provides a detailed and principled motivation from information-bottleneck in contrastive learning. It makes sense that 1) the semantics of time series in the context of contrastive learning (leveraging the MI between real and augmented samples) lies more in the time-domain as it can separately record the fluctuation and trend for different frequencies; and 2) the prototype refinement in time-domain should be adjusted to reflect the fundamental discrimination in the contrastive learning process. The experiments are conducted properly and extensively.

2. Presentations: The diagrammatic presentation of methods and the mathematical interpretations are well structured. In particular, the MI interpretation justifies the augmentation method can avoid incomplete semantics.

3. Significance: Data augmentation of time series, due to the ambiguity of “semantics” label, is a long-lasting direction to pursue. Tackling this problem is a step towards unsupervised learning for TS.

4. Originality: To my best knowledge, prototype refinement in the learning contrastive learning process, especially for time series, is a quite unique direction. Also, in time series analysis, the prototype-guided semantic data augmentation is also an innovation as phase alignment and amplitude noise are both concerned.

---

> ### Author Rebuttal · Authors · 2026-03-31
>
> We would like to thank the reviewer for the comments. Due to the response-length limit, we summarize only the key points here, with supporting figures/tables provided at https://anonymous.4open.science/r/ICML26-B2FC/Results.pdf.
>
> ### **Comment 1: Task-dependent requirements for data augmentation and contrastive learning**
>
> **Response:**
> We agree that forecasting and classification tasks place different demands on data augmentation and representation learning. For forecasting, the learned representation should preserve temporal continuity, predictive dependencies, trends and periodic dynamics, since overly aggressive perturbations may damage the structures needed for predicting the future. While for classification, the representation should preserve discriminative morphology while remaining robust to the local noise and non-essential variations.
>
> Our method is not designed as a task-specific augmentor for only one downstream objective. Instead, it follows a more general semantics-aware principle: the prototype-guided alignment first identifies regions that are more consistent with the current semantic anchor, and then differentiated transformations are applied to different regions. The core idea is to minimize unnecessary corruption of semantically important segments while allowing stronger perturbations on weakly related regions to improve view diversity and robustness.
>
> Under this unified principle, the same framework naturally serves different tasks with different emphases. For forecasting, it preserves predictive dynamics. For classification, it suppresses nuisance variations that interfere with discriminative representations. In this sense, the semantic-preserving property in our method is not tied to a specific downstream task, but is defined through a prototype consistency, whose benefit manifests differently across tasks.
>
> ### **Comment 2: Scalability w.r.t. data size**
>
> **Response:**
> We agree that the scalability is important, especially for time-series methods involving prototypes and alignment. In our method, the additional computation is introduced mainly during self-supervised pre-training, specifically by prototype-guided segmentation and prototype refinement, which does not introduce an extra cost for the downstream inference. Thus, the main overhead lies in training rather than the deployment.
>
> To directly evaluate scalability w.r.t. the data size, we vary the training-set ratio while keeping the other settings fixed. As shown in **Table A1**, the average training time per epoch increases from **228 ms** at **25\%** data to **421 ms** at **50\%**, **672 ms** at **75\%**, and **848 ms** at **100\%**. This shows an approximately linear growth trend w.r.t. the amount of training data.
>
> We also examine scalability w.r.t. the sequence length. As shown in **Table A2**, when the input length increases from **64** to **128**, **256**, and **512**, the average training time per epoch increases from **688 ms** to **780 ms**, **848 ms**, and **934 ms**, respectively. The increase is moderate rather than dramatic. Our interpretation is that longer sequences do increase the cost of prototype matching and DTW-based alignment, but this growth remains limited in practice, because the prototype-matching module itself accounts for only a relatively small fraction of the total training cost.
>
> In addition, we compare the practical training time of ProSAR with three representative baselines over 20 epochs in **Figure A1**. The average epoch time is **0.848 s** for ProSAR, compared with **0.583 s** for TimesURL [A1], **1.296 s** for FreRA [A2], and **1.552 s** for AutoTCL [A3]. Thus, though ProSAR introduces additional prototype-guided operations, its practical runtime remains moderate and is still lower than several strong baselines.
>
> This interpretation is also consistent with our module-level runtime profiling. In the current implementation, the dominant cost still comes from the main optimization stage (**loss + backward = 81.48\%** of the epoch time), whereas the DTW-related overhead remains relatively small (**DTW alignment + matching = 2.87\%**). Thus, even as the data size or sequence length increases, the prototype-guided alignment does not become the dominant computational bottleneck.
>
> [A1] TimesURl: Self-supervised contrastive learning for universal time series representation learning, AAAI 2024.
>
> [A2] FreRA: a frequency-refined augmentation for contrastive learning on time series classification, ACM SIGKDD 2025.
>
> [A3] Parametric augmentation for time series contrastive learning, ICLR 2024.

---

### Decision · Program_Chairs · 2026-04-30

**Decision:**

Accept (regular)

**Comment:**

This paper introduces a semantic-aware time series contrastive learning approach aimed at improving interpretability and controllability. Although most reviewers recognize the methodological rigor and originality of the work, there remain concerns regarding the scope of experimental validation, especially in transfer learning and semi-supervised contexts. To strengthen the contribution, the authors are advised to enhance both their empirical assessments and the manuscript's overall presentation.